# Building Cooperative Embodied Agents Modularly with Large Language Models

**Hongxin Zhang**[1*], **Weihua Du**[2*], **Jiaming Shan**[3], **Qinhong Zhou**[1]
**Yilun Du**[4], **Joshua B. Tenenbaum**[4], **Tianmin Shu**[4], **Chuang Gan**[1,5]
[1]University of Massachusetts Amherst,
[2]Institute for Interdisciplinary Information Sciences, Tsinghua University,
[3]Shanghai Jiao Tong University, [4]MIT, [5]MIT-IBM Watson AI Lab

## Abstract

In this work, we address challenging multi-agent cooperation problems with decentralized control, raw sensory observations, costly communication, and multi-objective tasks instantiated in various embodied environments. While previous research either presupposes a cost-free communication channel or relies on a centralized controller with shared observations, we harness the commonsense knowledge, reasoning ability, language comprehension, and text generation prowess of LLMs and seamlessly incorporate them into a cognitive-inspired modular framework that integrates with perception, memory, and execution. Thus building a **Co**operative **E**mbodied **L**anguage **A**gent *CoELA*, who can plan, communicate, and cooperate with others to accomplish long-horizon tasks efficiently. Our experiments on C-WAH and TDW-MAT demonstrate that *CoELA* driven by GPT-4 can surpass strong planning-based methods and exhibit emergent effective communication. Though current Open LMs like LLAMA-2 still underperform, we fine-tune a *CoLLAMA* with data collected with our agents and show how they can achieve promising performance. We also conducted a user study for human-agent interaction and discovered that *CoELA* communicating in natural language can earn more trust and cooperate more effectively with humans. Our research underscores the potential of LLMs for future research in multi-agent cooperation. Videos can be found on the project website https://vis-www.cs.umass.edu/Co-LLM-Agents/.

## 1 Introduction

Humans are adept at cooperating and communicating with others when solving complex tasks (Woolley et al., 2010). Building embodied agents that can also engage in and assist humans in everyday life is a valuable but challenging task, considering the complexity of perception, partial observation, long-horizon planning, natural language communication, and so on (Deitke et al., 2022).

Large Language Models (LLMs) have exhibited remarkable capabilities across various domains, implying their mastery of natural language understanding, dialogue generation, rich world knowledge, and complex reasoning capability (OpenAI, 2023; Touvron et al., 2023; Brown et al., 2020; Bubeck et al., 2023). Recent research has also demonstrated that LLMs can drive embodied agents for single-agent tasks through zero-shot prompting for instruction following tasks (Huang et al., 2022a) or few-shot prompting for more complex long-horizon tasks (Song et al., 2022). However, building cooperative embodied agents to work with other agents or with humans under decentralized settings with costly communication remains challenging and rarely explored, where they also need to have strong abilities for cooperative planning and efficient communication. To date, it still remains unclear whether LLMs have such abilities necessary for distributed embodied multi-agent cooperation.

Therefore, this paper aims to investigate how to leverage LLMs to build cooperative embodied agents that can collaborate and efficiently communicate with other agents and humans to accomplish long-horizon multi-objective tasks in a challenging decentralized setting with costly communication. To this end, we focus on an embodied multi-agent setting as shown in Figure 1, where two **decentralized**

---

* denotes equal contribution.

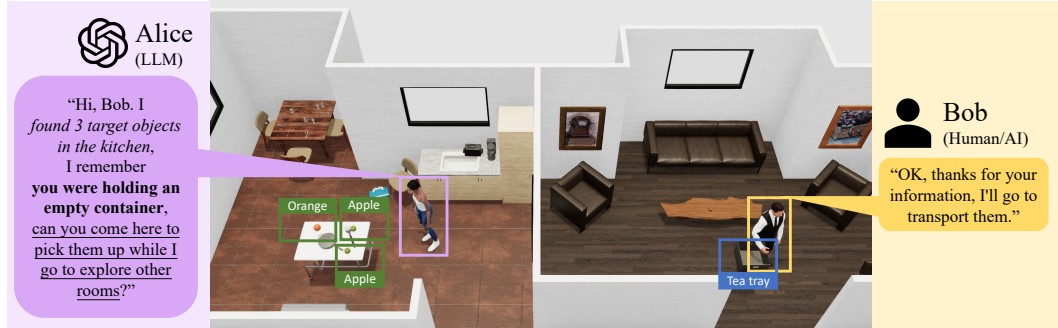

Figure 1: A challenging multi-agent cooperation problem with decentralized control, raw sensory observations, costly communication, and long-horizon multi-objective tasks.

embodied agents have to cooperate to finish a **multi-objective** household task efficiently with **complex partial observation** given. Specifically, **communication in our setting takes time** as in real life, so the agents can't simply keep free talking with each other. To succeed in this setting, agents must i) perceive the observation to extract useful information, ii) maintain their memory about the world, the task, and the others, iii) decide what and when to communicate for the best efficiency and iv) plan collaboratively to reach the common goal.

Inspired by prior work in cognitive architectures (Laird, 2019), we present *CoELA*, a **Co**operative **E**mbodied **L**anguage **A**gent, a cognitive architecture with a novel modular framework that utilizes the rich world knowledge, strong reasoning ability and mastery natural language understanding and generation capability of LLMs, who plan and communicate with others to cooperatively solve complex embodied tasks. Our framework consists of five modules, each to address a critical aspect of successful multi-agent cooperation, including a Perception Module to perceive the observation and extract useful information, a Memory Module mimicking human's long-term memory to maintain the agent's understanding of both the physical environment and other agents, a Communication Module to decide *what* to communicate utilizing the strong dialogue generation and understanding capability of LLMs, a Planning Module to decide high-level plans including *when* to communicate considering all the information available, and an Execution Module to execute the plan by generating primitive actions using procedures stored in the memory module.

We instantiate our challenging setting and evaluate our framework on two embodied environments: ThreeDWorld Multi-Agent Transport (TDW-MAT) and Communicative Watch-And-Help (C-WAH). Our experimental results indicate that *CoELA* can perceive complex observations, reason about the world and others' state, communicate efficiently, and make long-horizon plans accordingly, as showcased in Figure 1 where *CoELA* divide the labor with its partner through natural language communication effectively. In particular, *CoELA* driven by GPT-4 can outperform strong planning-based baselines by achieving more than 40% efficiency improvements and exhibiting emergent efficient communication. Though Open LMs like LLAMA-2 still underperform, we utilize parameter-efficient fine-tuning techniques LoRA (Hu et al., 2021) to train a *CoLLAMA* on few data collected with our agents and gain promising performance. In the user study, we also discover that *CoELA* communicating with humans in natural language can earn more trust. Our contribution includes:

- We formalized a challenging multi-agent embodied cooperation problem with decentralized control, complex partial observation, costly communication, and long-horizon multi-objective tasks, and instantiated it in two embodied environments: C-WAH and TDW-MAT.
- We presented a novel cognitive-inspired modular framework that utilizes the strong planning and communication capability of the LLMs to build cooperative embodied agents *CoELA*, surpassing strong planning-based methods.
- We conducted a user study to evaluate the possibility of achieving effective and trustworthy human-AI cooperation using LLMs.

## 2 RELATED WORK

**Multi-Agent Cooperation and Communication** The field of multi-agent cooperation and communication has a long-standing history (Stone & Veloso, 2000). Many platforms have been proposed for various multi-agent tasks (Lowe et al., 2017; Resnick et al., 2018; Shu & Tian, 2018; Jaderberg

et al., 2019; Samvelyan et al., 2019; Suarez et al., 2019; Baker et al., 2019; Bard et al., 2020). Other works focused on methods that improves communication efficiency (Jiang & Lu, 2018; Das et al., 2019; Wang et al., 2021; Wan et al., 2022), cooperation in visually rich domains (Jain et al., 2020), or grounding communications in environments (Patel et al., 2021; Mandi et al., 2023; Narayan-Chen et al., 2019). For embodied intelligence, Puig et al. (2021; 2023) explored the social perception of the agents during their cooperation. However, these platforms either neglects communication (Jaderberg et al., 2019; Samvelyan et al., 2019; Carroll et al., 2019; Puig et al., 2021; 2023), or use uninterpretable continuous vectors (Jiang & Lu, 2018; Das et al., 2019) or limited discrete symbols (Lowe et al., 2017; Jaques et al., 2019; Jain et al., 2020; Patel et al., 2021; Resnick et al., 2018) for communication. In contrast, we propose a more challenging setting where no presupposed free communication channel exists, and **distributed** agents need to use natural language to communicate *efficiently* with others, especially humans.

**Language Agents** Recently, numerous studies have explored *language agents* which use LLMs for sequential decision-making (Yang et al., 2023; Wang et al., 2023b; Xi et al., 2023; Sumers et al., 2023). Although LLMs still face challenges when solving complex reasoning problems (Bubeck et al., 2023), a substantial body of work demonstrates their capacity to make plans (Sharma et al., 2021; Raman et al., 2022; Pallagani et al., 2022; Gramopadhye & Szafir, 2022; Yuan et al., 2023; Li et al., 2022; Wang et al., 2023d), especially in embodied environments (Li et al., 2023a; Padmakumar et al., 2022; Kolve et al., 2017; Shridhar et al., 2020; Misra et al., 2018; Zhu et al., 2017; Brodeur et al., 2017; Xia et al., 2018; Savva et al., 2019; Xiang et al., 2020; Jain et al., 2020; 2019). Specifically, Liang et al. (2022); Song et al. (2022) used codes or few-shot prompting to directly generate plans, Huang et al. (2022b) built an inner monologue with environment feedback to improve planning, Ahn et al. (2022) combined robotic affordances and LLMs for grounded instruction following. There has also been a line of work utilizing multiple LLMs to cooperate or debate with each other "in mind" to strengthen the single agent's capability to solve complex tasks (Li et al., 2023b; Du et al., 2023; Wang et al., 2023c), different from their "free self-talk" setting, our decentralized language agents must plan about when and what to communicate carefully since it's costly in real-life. More recently, Park et al. (2023) built an agent society using LLMs augmented with memories to simulate human behavior. In contrast to the above, our work addresses a more *challenging* multi-agent cooperation problem, characterized by decentralized control, complex observations, **costly communication**, and **long-horizon multi-objective tasks**. We also study the capability of Open LMs like LLAMA-2 and tine-tune a *CoLLAMA* using LoRA with data collected by our agents in embodied environments to demonstrate their promising performance for building better cooperative embodied agents.

## 3 COOPERATIVE PLANNING UNDER DEC-POMDP-COM

Our setting can be defined as an extension of the decentralized partially observable Markov decision process (DEC-POMDP) (Bernstein et al., 2002; Spaan et al., 2006; Goldman & Zilberstein, 2003), which can be formalized by $(n, S, \{\Sigma_i\}, \{A_i\}, \{O_i\}, T, G, R, \gamma, h)$, where $n$ denotes the number of agents; $S$ is a finite set of states; $A_i = A_i^W \cup A_i^C$ is the action set for agent $i$, including a finite set of world actions $A_i^W$ and a communication action $A_i^C$ to send a message $\sigma_i \in \Sigma_i$; $O_i = O_i^W \times O_i^C$ is the observation set for agent $i$, including world observations $O_i^W$ the agent receives through its sensors, and $O_i^C = \Sigma_1 \times \cdots \times \Sigma_n$ the set of possible messages the agent can receive from any of its teammates; $T(s, a, s') = p(s'|s, a)$ is the joint transition model which defines the probability that after taking joint action $a \in A_1 \times \cdots \times A_n$ in $s \in S$, the new state $s' \in S$ is achieved; $G = \{g_1, \cdots, g_k\}$ defines the task with several sub-goals for the agents to finish; $R(s, a, s') = -c(a) + \sum_{i=1}^{k} \mathbb{1}(s' = g_i) - \mathbb{1}(s = g_i)$ is the reward function to the team, where $c(a)$ is the cost for action $a$, and $\mathbb{1}(\cdot)$ checks if the sub-goal $g_i$ is satisfied in the world state $s$; $\gamma$ is the discount rate and $h$ is the planning horizon. In the remainder of this paper, we focus on noise-free broadcast communication and limit our discussion to two agents, though our methods and experiments are generalizable to more than two agents.

We instantiate the problem with two decentralized intelligent embodied agents (including humans) cooperating to accomplish a long-horizon rearrangement task (Batra et al., 2020) in an indoor multi-room environment. The agents are capable of executing one of the actions from the action space $\mathcal{A} = \mathcal{A}_{\text{NAV}} \cup \mathcal{A}_{\text{INT}} \cup \mathcal{A}_{\text{COM}}$, where $\mathcal{A}_{\text{NAV}}$ includes navigation actions, $\mathcal{A}_{\text{INT}}$ includes interaction actions and $\mathcal{A}_{\text{COM}}$ includes a communication action with which the agent can send a message in natural language to broadcast to others. The rearrangement task is defined with several predicates

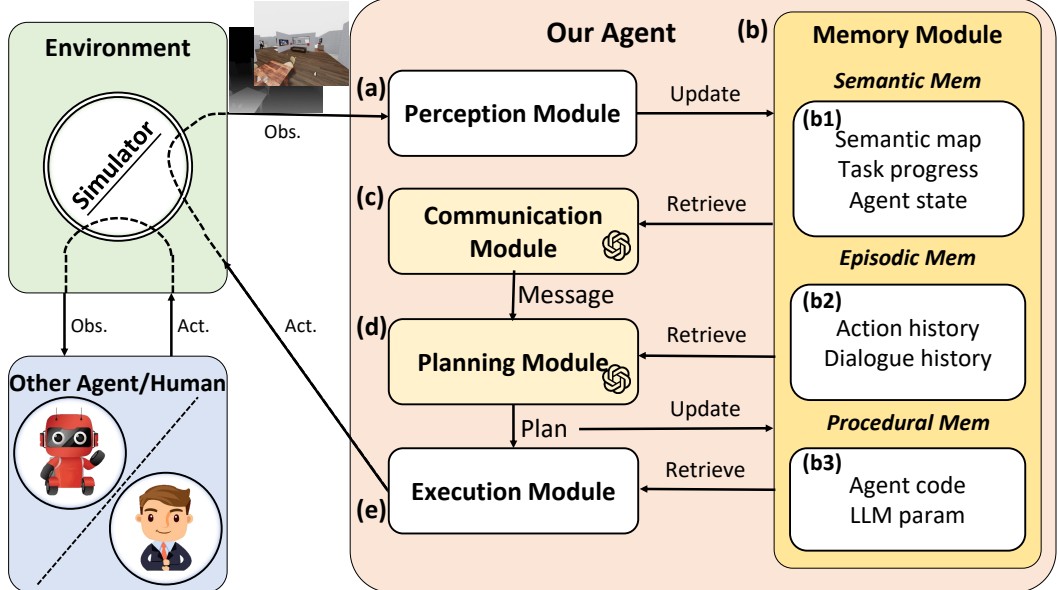

Figure 2: An overview of *CoELA*. There are five key modules in our framework: (c) The Communication Module and (d) the Planning Module leverage LLMs to generate messages and make plans, (b) The Memory Module stores the agent's knowledge and experience about the world and others in semantic, episodic and procedural memory respectively, (a) The Perception Module and (e) the Execution Module interact directly with the external environment by perceiving raw observations and generating primitive actions. More design details can be found in Appendix A.

$g_i$ with counts to be satisfied, such as `ON(plate,dinnertable):2` representing a sub-task of putting two plates onto the dinner table.

## 4 BUILDING COOPERATIVE EMBODIED AGENTS MODULARLY WITH LLMS

### 4.1 FRAMEWORK OVERVIEW

Inspired by the cognitive architectures (Langley et al., 2009; Laird, 2019; 2022), we build *CoELA*, a Cooperative Embodied Language Agent with novel modular framework integrating the strong reasoning ability and language generation capability of LLMs. As shown in Figure 2, *CoELA* consists of five key modules: (a) Perception, (b) Memory, (c) Communication, (d) Planning, and (e) Execution. At each interaction step, *CoELA* first uses (a) Perception Module to perceive the raw sensory observation received from the environment, then updates the (b) Memory Module with extracted new information, which stores its knowledge and experience of the world and others. *CoELA* tackles the challenge of efficient communication with a two-step method: first decide on *what* to send, then decide *whether* to send this message or choose another plan by deliberately using (c) The *Communication Module* to retrieve related information from (b) and utilize an LLM to generate the best message to send "in mind" beforehand, then leverages (d) the *Planning Module* driven by LLM with strong reasoning ability to make the decision on which plan to take given the related information retrieved from (b) and available actions proposed regarding the current state. The generated plan is then used to update (b2) the Episodic Memory. Finally, (e) the *Execution Module* retrieves procedural knowledge stored in (b3) to turn the high-level plan into primitive actions executable in the environment.

### 4.2 PERCEPTION MODULE

For embodied agents to be helpful in the real world, they have to perceive raw observations gained through sensors and extract useful information for downstream higher-order reasoning. We incorporate the Perception Module to deal directly with the complex visual observation received from the environment by training a Mask-RCNN (He et al., 2017) to predict the segmentation masks from the RGB image, then build 3D point clouds using the RGB-D image, extract useful high-level information such as the states of the key objects and build a local semantic map.

### 4.3 MEMORY MODULE

It's of vital importance for an agent to maintain a memory of the knowledge and experience it has of the world and others, we mimic human's long-term memory (Atkinson & Shiffrin, 1968; Wang &

Laird, 2006; Nuxoll & Laird, 2012) and design Semantic memory, Episodic Memory, and Procedural Memory for *CoELA*.

**Semantic Memory** stores *CoELA*'s knowledge about the world including a semantic map, the task progress, the state of self, and the state of others. Each time a new observation is received and perceived by the Perception Model, the Semantic Memory is updated accordingly. To be noticed, *CoELA*'s knowledge about the world may not be accurate since other agents may interact with the objects and change their states without its awareness. Dealing with imparities between the memory and the description of the world from others adds even more challenges.

**Episodic Memory** stores *CoELA*'s experience about the past including the action history and dialogue history. Each time *CoELA* executes a new action including sending out a message or receiving a new message, the related information is added to the Episodic Memory.

**Procedural Memory** contains knowledge including how to carry out specific high-level plans in a specific environment implemented in code and the neural models' parameters.

### 4.4 COMMUNICATION MODULE

To deal with the *what* to send problem, we deliberately design a Communication Module utilizing the strong free-form language generation capability of the LLMs to act as a message generator. To better condition the LLMs on the cooperative task and avoid inefficient casual chatting, the Communication Module first retrieves the related information from the Memory Module including the semantic map, task progress, agent state, others state, and the action and dialogue history, then convert these into text descriptions using templates, finally prompt the LLMs with the concatenation of *Instruction Head, Goal Description, State Description, Action History,* and *Dialogue History* to generate the message to send. To better constrain LLMs' generated messages, a note at the end of the prompt is added and two seed messages are appended at the beginning of the Dialogue History to elicit deserved effective communication behavior. Detailed prompt design in Appendix. A.3.

### 4.5 PLANNING MODULE

*CoELA* needs a strong Planning Module to make decisions on which action to take utilizing all available information gathered and stored so far to maximize cooperation efficiency. While designing such a module from scratch consumes large human expert efforts and is nearly impossible to generalize, we utilize powerful LLMs directly as the Planning Module by first retrieving the related information from the Memory Module and converting them into text descriptions as in the Communication Module, then compile an Action List of all available high-level plans proposed according to the current state and the procedural knowledge stored for the LLMs to make the choice, which formalization makes it easier for the LLMs to concentrate on the reasoning and make an executable plan without any few-shot demonstrations easily, finally prompting the LLMs with current information and the proposed Action List to generate a high-level plan. We also use the zero-shot chain-of-thought prompting technique introduced by Kojima et al. (2022) to encourage the LLMs to carry out more reasoning before giving the final answer. More details can be found in Appendeix. A.4.

### 4.6 EXECUTION MODULE

As shown in (Deitke et al., 2022), solving challenging embodied tasks requires modular methods to tackle the complexity of tasks. We found that while LLMs were effective at making high-level plans, they were poor at making low-level controls, as also discussed in (Wu et al., 2023). Thus, to enable effective and generalized cooperation decision-making in different environments, we design an Execution Module to generate primitive actions to execute a given high-level plan robustly in a specific environment, allowing the Planning Module to be generalizable and focus more on solving the overall task with LLMs' rich world knowledge and strong reasoning ability. Practically, this design can also reduce the LLM inference time and is time-saving and economical. *CoELA* retrieves the procedures in its Memory Module regarding the plan generated by the Planning Module and then carries out the procedure with primitive actions suitable for the environment.

## 5 EXPERIMENTS

### 5.1 EXPERIMENTAL SETUP

**ThreeDWorld Multi-Agent Transport (TDW-MAT)** is a multi-agent embodied task extended from the ThreeDWorld Transport Challenge (Gan et al., 2022) with more types of objects and containers, more realistic object placements, and communication between agents supported, built on top of the TDW platform (Gan et al., 2021), which is a general-purpose virtual world simulation platform. The agents are tasked to transport as many target objects as possible to the goal position with the help

of containers as tools. The agents receive ego-centric 512×512 RGB-D images as observation and have an action space of low-level navigation control, interaction, and communication. We selected 6 scenes from the TDW-House dataset and sampled 2 out of the two types of tasks *food* and *stuff* in each of the scenes, making a test set of 24 episodes, and instantiate the horizon $h$ with 3000 frames.

**Communicative Watch-And-Help (C-WAH)** is extended from the Watch-And-Help Challenge (Puig et al., 2021) built on a realistic multi-agent simulation platform, VirtualHome-Social (Puig et al., 2018; 2021), where we focus more on cooperation ability and support communication between agents. We conduct experiments under both symbolic and visual observation settings. The task is defined as five types of common household activities and represented as various predicates with counts to be satisfied. We sampled 2 tasks from each of the five types of activities to construct a test set of 10 episodes and instantiate the horizon $h$ with 250 steps. More details can be found at Appendix. B.

**Metrics** We use the *Transport Rate (TR)*, the fraction of the sub-goals satisfied on TDW-MAT, and the *Average Steps L* taken to finish the task on C-WAH as main efficiency metrics respectively and calculate *Efficiency Improvement (EI)* of cooperating with other agents as $\Delta M/M_0$, where $\Delta M$ denotes the main efficiency metric difference, and $M_0$ denotes the larger one of the main efficiency metric for numerical stability.

## 5.2 BASELINES

**MCTS-based Hierarchical Planner(MHP)** is adopted from the strongest baseline in the original Watch-And-Help Challenge, which is a Hierarchical Planner with a high-level planner based on MCTS and a low-level planner based on regression planning (Korf, 1987).

**Rule-based Hierarchical Planner(RHP)** is adopted from the strong performing baseline in the original ThreeDWorld Transport Challenge, which is a Hierarchical Planner with a high-level planner based on heuristics rules and a low-level A-start-based planner to navigate with semantic map, using Frontier Exploration strategy which randomly samples a way-point from an unexplored area as a sub-goal for exploration.

**Multi-Agent Transformer(MAT)** is a MARL baseline that applies a centralized decision transformer to generate actions from shared observations (Wen et al., 2022). To apply MAT in our setting, we make the compromise to feed the oracle semantic map and the agent states as observation and stack up to 50 frames as an RL step since TDW-MAT is too hard for it with long-horizon and sparse reward signals. We train MAT on the training set with more details in Appendix. C.1.

**Implementation Details.** We train a Mask-RCNN on the training set for the Perception Module and instantiate *CoELA* with the most powerful LLM GPT-4 from the OpenAI API[1] with the default parameter of temperature 0.7, top-p 1, and max tokens 256 unless other stated. We also conduct experiments with Open LLM LLAMA-2-13b-chat (Touvron et al., 2023) and fine-tune a *CoLLAMA* with LoRA (Hu et al., 2021) on a small set of human-filtered high-quality trajectory data collected with our agents. More details are deferred to the Appendix. C.3.

## 5.3 RESULTS

### 5.3.1 COLLABORATING WITH AI AGENTS

***CoELA* cooperates better with baseline agent**
As shown in Table 1, compared with RHP doing the task alone, cooperating with *CoELA* leads to a higher TR and EI than cooperating with another RHP (0.69(36%) v.s. 0.61(29%)), even without any knowledge of the inner working mechanism of others, showing *CoELA* can reason about the other agent's state well without hand-designed heuristics. From Table 2, we can observe the same performance boost of cooperating with *CoELA* on C-WAH of 45% compared to 33% of cooperating with the same MHP.

| | Symbolic Obs | Visual Obs |
|---|---|---|
| **MHP** | 111 | 141 |
| **MHP + MHP** | 75(↑33%) | 103(↑26%) |
| **MHP + *CoELA*** | 59(↑45%) | 94(↑**34%**) |
| ***CoELA* + *CoELA*** | **57(↑49%)** | **92(↑34%)** |

Table 2: **Quantitative results on C-WAH.** We report the average steps(Efficiency Improvement) here over 5 runs for MHP and 1 run for *CoELA* due to cost constraints. The best performance is achieved when cooperating with *CoELA*.

***CoLLAMA* is in competence with GPT-4 to drive *CoELA*** Two *CoELA* cooperate together can further boost the TR to 0.71 and 0.85 on TDW-

---

[1]Our main experiments are done between 2023.9.1-2023.9.28 and 2023.5.1-2023.5.16

| | RHP | RHP + RHP | RHP + *CoELA* | *CoELA + CoELA* | | | MAT* |
| --- | --- | --- | --- | --- | --- | --- | --- |
| | | | | GPT-4 | LLAMA-2 | CoLLAMA-2 | |
| TDW-MAT | | | | | | | |
| *Food* | 0.49 | 0.67(↑25%) | 0.79(↑**39%**) | **0.82**(↑38%) | 0.57(↑9%) | 0.73(↑33%) | / |
| *Stuff* | 0.36 | 0.54(↑34%) | 0.59(↑34%) | 0.61(↑41%) | 0.48(↑11%) | **0.66**(↑**44%**) | / |
| *Total* | 0.43 | 0.61(↑29%) | 0.69(↑36%) | **0.71**(↑**39%**) | 0.53(↑10%) | 0.70(↑38%) | / |
| TDW-MAT w/ Oracle Perception | | | | | | | |
| *Food* | 0.52 | 0.76(↑33%) | 0.85(↑40%) | **0.87**(↑**41%**) | 0.60(↓3%) | 0.78(↑34%) | 0.13(↓) |
| *Stuff* | 0.49 | 0.74(↑34%) | 0.77(↑35%) | **0.83**(↑**41%**) | 0.63(↑19%) | 0.81(↑38%) | 0.17(↓) |
| *Total* | 0.50 | 0.75(↑34%) | 0.81(↑37%) | **0.85**(↑**41%**) | 0.62(↑8%) | 0.80(↑36%) | 0.15(↓) |

Table 1: **Quantitative results on TDW-MAT.** We report the average *Transport Rate*(*Efficiency Improvement*) here over 5 runs for RHP and 1 run for *CoELA* due to cost constraints. *MAT uses central observation and oracle perception. The best results are in **bold**. The best performance is achieved when cooperating with *CoELA*.

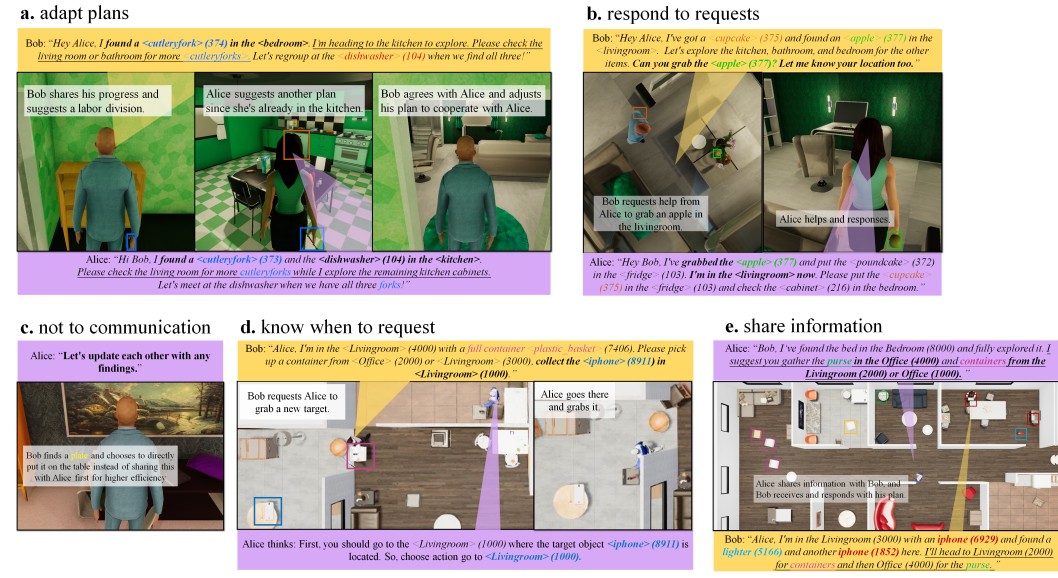

Figure 3: **Example cooperative behaviors** demonstrating *CoELA* can communicate effectively and are good cooperators.

MAT without and with Oracle Perception. While replacing GPT-4 with open Model LLAMA-2 leads to a significant performance drop, our fine-tuned *CoLLAMA* can gain a competitive performance of 0.70 TR and even surpass GPT-4 on the subtask of *Stuff* where GPT-4 performs not so well, showing the promising future of fine-tuning open LLMs with our proposed framework on embodied environments for even better cooperative embodied agents.

***CoELA* exhibit efficient communication and effective cooperation behavior** To better understand the essential factors for effective cooperation, we conduct a qualitative analysis of the agents' behaviors exhibited in our experiments and identified several cooperative behaviors: *CoELA* **share** progress and information with others, know when to **request** help and can **respond** to others' requests, can **adapt** plans considering others and knows **when not to** communicate, as shown in Figure 3. We discuss some here and the remaining in the Appendix. C.4.

### 5.3.2 COLLABORATING WITH HUMANS

It's our ultimate goal to build agents that can cooperate with humans, a user study is important. We conducted human experiments on the C-WAH where the agent Alice is controlled by real humans.

We recruited 8 human subjects to perform the experiments under four scenarios: cooperating with the **MHP**[2], *CoELA*, *CoELA* **w/o communication**, and doing the task alone. Subjects have access to the same observation and action space as the agents, they can click on visible objects and select actions

---

[2]A template communication is used here to study humans' communication preference, details in Appendix F

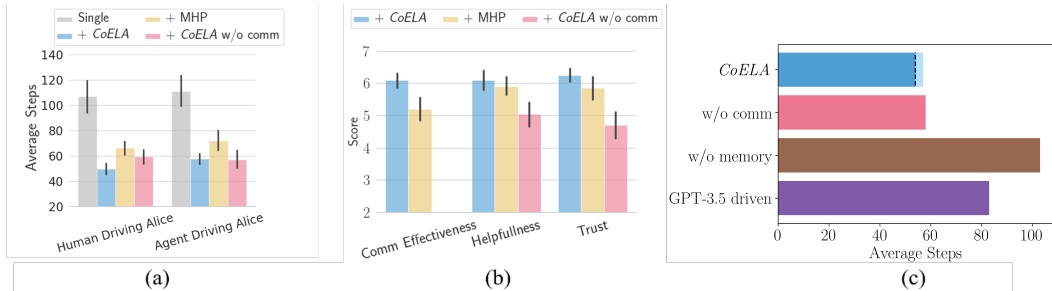

Figure 4: **Human experiments results** (a) The Average steps when collaborating with Humans and agents. (b) Subjective Rating Humans give when cooperating with different agents. Humans trust *CoELA* communicating in natural language more and cooperate more efficiently with them. **Ablation results** (c) The light-colored portions represent the number of steps used for communication. The Memory Module and a strong LLM for the Planning Module are important, while the Communication Module matters more when cooperating with humans.

to interact with them, including navigation to each room and communication through a chat box. We gave each subject a tutorial and they had the chance to get familiar with the interface in a few pilot trials. We evaluate the same 10 tasks as in previous experiments and each task was performed by at least 2 subjects, making 80 trials in total. We made sure each subject do 10 trials with at least two trials under each scenario. After each trial including a baseline to cooperate with, we asked subjects to rate the agent they just cooperated with on a 7-point Likert Scale based on three criteria adapted from Puig et al. (2021): (i) *How effective do you think of your communication with the other agent Bob? Did it understand your message and/or share useful information with you?* (ii) *How helpful do you find the other agent Bob? Did it help you achieve the goal faster?* (iii) *How much do you trust the other agent Bob? Would you feel safe doing the task with it, or you rather do the task alone?*

As we can see in Figure 4a, when cooperating with humans, *CoELA* still performs better than MHP, and when communication is unable, *CoELA* w/o communication encounters a performance drop. As reported in Figure 4b, we also observe that humans would trust the agents more if they can communicate with humans (trust score of 6.3 v.s. 4.7 for *CoELA* v.s *CoELA* w/o communication, p=0.0003 over the t-test), and therefore achieves better cooperation. Compared with MHP using template language to communicate, humans prefer to collaborate with *CoELA* who communicates in natural language and can understand and respond to Human dialogues. We show an effective communication example in Figure 10, where the human first shares his progress with *CoELA* and suggests a labor division, *CoELA* understands and responds with its future plan as well, resulting in a perfect division of the exploration trajectory. These results imply promising futures for leveraging LLMs to build cooperative embodied agents that can successfully work with humans.

## 5.4 ANALYSIS

**Do we need a strong LLM for the Planning and Communication Module?** As shown in Figure 4c, when we replace GPT-4 with GPT-3.5 to drive *CoELA*, the agents would need more steps to finish the task. GPT-3.5 makes more reasoning errors about the state and therefore generates more implausible plans, which leads *CoELA* to spend more time finishing the task. GPT-3.5 also tends to generate unuseful messages more often than GPT-4. The performance gap can be attributed to more advanced reasoning and Theory of Mind abilities of GPT-4, which is also observed by Bubeck et al. (2023).

**Is the communication effective?** Though communication still fails in some cases, as shown in Figure 3, our agent exhibits effective communication behaviors, such as sharing information, requesting help, responding to requests, and knowing when not to communicate. More importantly, natural language communication provides us with a lens to understand the decision-making of the agents and could lead to better cooperation between humans and AI (as shown in section 5.3.2). We did not observe a significant performance drop when disabling communication among AI agents (as shown in Figure 4c), because carrying out efficient communication in our setting is extremely challenging as communication costs time, requiring agents to model others accurately and understand the ambiguity of the natural language itself, which current LLMs still can not master robustly.

**Is the Memory Module and Execution Module effective?** As shown in Figure 4c, the steps needed to finish the task for the agent with no Memory Module nearly double, showing the importance of the Memory Module to store and update the knowledge and experience of the scene and the others. We also tried to remove the Execution Module and let the Planning Module make low-level control

directly at every step. However, this slows down the inference process largely and all our trials perform poorly and struggle to finish any task.

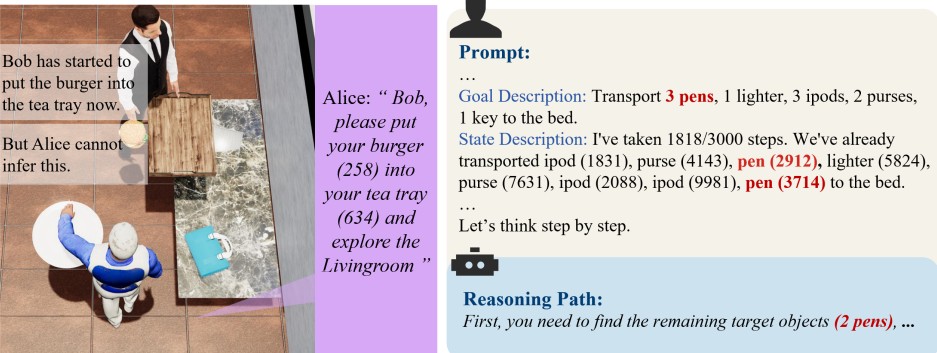

Figure 5: **Failure cases on TDW-MAT**. (a) The Agent fails to reason the other one is already putting the burger into the container. (b) The LLM counts the number of the remaining target objects wrong.

## 5.5 FAILURE CASES AND LIMITATIONS OF LLM

Though *CoELA* built with sota LLMs is effective and has achieved impressive results, we find that the agent still falls short in several essential capabilities. We provide an in-depth analysis of its limitations and share some insights on designing better cooperative embodied agents for future work.

**Limited usage of 3D spatial information.** *CoELA* did not incorporate the spatial information of objects and rooms into consideration due to the challenge of effectively introducing the spatial information to pure text language models. This may cause the agents to come up with a semantic sound exploration plan which is actually time-consuming. Work on multi-modal large models capable of both processing visual modalities effectively and generating natural language fluently (Huang et al., 2023; Driess et al., 2023; Lu et al., 2022) would help overcome this limitation and build better grounded embodied agents.

**Lack of effective reasoning on low-level actions.** To help LLMs better focus on solving the overall task, we abstract high-level plans for LLMs to directly reason on, reducing the potential decision space significantly, but also making it unaware of the execution of low-level actions, and impossible to reason over them, which may lead to plausible but ineffective decisions. For example in Figure 5a, Alice saw Bob holding a container and a target object in both hands and figured he may not know how to utilize the containers, so sent a message to instruct him to put the object into the container, though Bob was actually putting in the objects at the same time, which is impossible for Alice to reason over now. Developing agents that can directly make low-level controls is essential for building better cooperative agents.

**Unstable performance on complex reasoning.** Although LLMs make correct reasoning most of the time, they still occasionally make mistakes, including misunderstanding the environment rules specified in the prompt, and incorrect reasoning over the number of unsatisfied goals (Figure 5b). These mistakes can cause failures in planning. This calls for developing LLMs with stronger instruction following and reasoning capability.

## 6 CONCLUSION

In this work, we propose a novel modular framework integrating the Large Language Models to build cooperative embodied agents *CoELA*, who can plan, communicate, and collaborate efficiently with other agents and humans in a challenging multi-agent setting with decentralized control, complex partial observation, costly communication, and multi-objective long-horizon tasks. Our experiments on two extended embodied multi-agent environments show the effectiveness of our proposed framework and exhibit several cooperative behaviors. We fine-tune a *CoLLAMA* from LLAMA-2 using data collected with our agents in embodied environments and showcase its promising performance to build better cooperative embodied agents. We also discover that *CoELA* communicating in natural language can cooperate better with humans and earn more trust from them. We believe that our work indicates promising future avenues to design even stronger embodied agents with LLMs for multi-agent cooperation. We further perform an in-depth analysis of the limitations of the current LLMs and highlight several potential solutions for building better embodied cooperative agents for the future.

## ACKNOWLEDGEMENT

We thank Zishuo Zheng and Zhiqing Sun for their insightful discussions and help with the experiments, Jeremy Schwartz and Esther Alter for setting up the ThreeDWorld environments. We thank the anonymous reviewers for their helpful suggestions. This work is funded in part by grants from ONR Science of AI Program, Google, Amazon, Cisco, Toyota Motor North America, and Mitsubishi Electric Research Laboratories.

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

# A  ADDITIONAL DETAILS ON THE FRAMEWORK

## A.1  PERCEPTION MODULE

To deal with raw sensory observations, a well-constructed Perception Module is needed for embodied agents to extract useful information for downstream higher-order reasoning.

In TDW-MAT, the environment provides an observation of $512 \times 512$ first-person view RGB image and Depth image. The agent first utilizes a pre-trained Mask-RCNN (He et al., 2017) to obtain the instance segmentation mask, then combines it with the depth image and the agent's position to project each pixel into the 3D world coordinate to obtain a 3D voxel semantic map, and finally accumulates along the height dimension to build a top-down 2D semantic map of size $L \times W \times 3$, where the first channel represents semantic classes including target objects, containers, destinations, and agents, and the last two channels represent the occupied and explored area respectively. Each element in the map denotes a grid of size $0.125m \times 0.125m$ in the scene. The agent also extracts the relationship of the objects with the help of instance segmentation masks and updates its Semantic Memory with the new information extracted from the observation.

To obtain a more suitable model for instance segmentation in a TDW simulation environment, we fine-tune the MASK-RCNN model pre-trained on the MS COCO dataset in training scenes. By random sampling in the training environments, we collected 53K $512 \times 512$ RGB images and obtained the ground truth instance segmentation mask from the environment as the training set. The fine-tuned model achieves 81.4% mAP@50 in the test set.

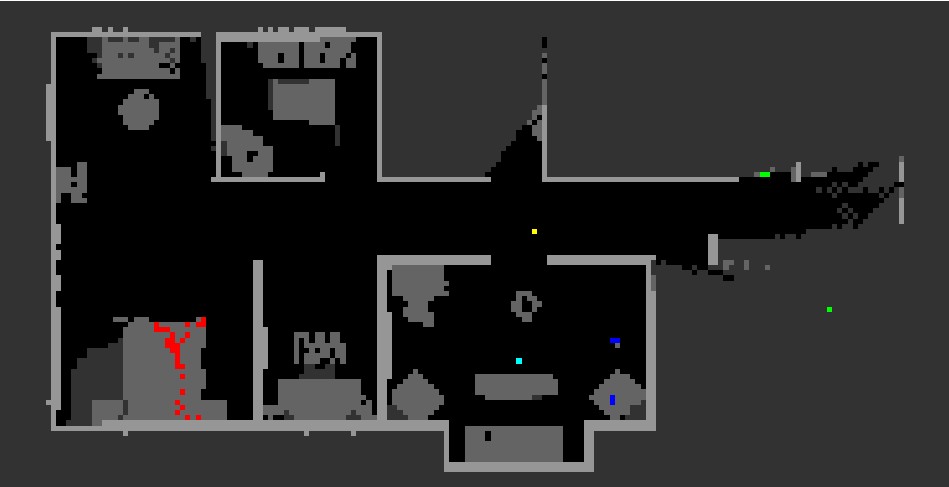

Figure 6: **A visualization of the semantic map stored in the Semantic Memory and updated with new observations at every time in the TDW-MAT environment.** The destination is shown in red, target objects are in blue, containers are in green, the agent is denoted with cyan, and the other agent's position in memory is denoted in yellow.

## A.2 MEMORY MODULE

We mimic human's long-term memory and design Semantic memory, Episodic Memory, and Procedural Memory for *CoELA* to store the knowledge and experience it has of the world, other agents, and itself.

**Semantic Memory** stores *CoELA*'s knowledge about the world including a semantic map as shown in Figure 6 built and updated with local map perceived from the Perception Module, the task progress which is initialized with all zeros and updated whenever the agent is in the range of the goal position, the state of self including positions, holding objects status, and the state of others in memory which is updated whenever the others is perceived in the observation. To be noticed, *CoELA*'s knowledge about the world may not be accurate since other agents may interact with the objects and change their states without its awareness. Dealing with imparities between the memory and the description of the world from others adds even more challenges.

**Episodic Memory** stores *CoELA*'s experience about the past including the action history and dialogue history. Each time *CoELA* executes a new action including sending out a message or receiving a new message, the related information is added to the Episodic Memory. Empirically, we only keep the last $K$ actions and $D$ dialogues for storage efficiency.

**Procedural Memory** contains knowledge including how to carry out specific high-level plans in a specific environment implemented in code and the neural models' parameters including LLMs and Mask-RCNN. In our current implementation, the Procedural Memory is never updated except for fine-tuning the model parameters, while it's interesting to design a learning mechanism for it as in (Wang et al., 2023a) as well.

## A.3 COMMUNICATION MODULE

It's important for cooperative embodied agents to be able to communicate effectively with others. Effective communication needs to solve two problems: *what* to send and *when* to send.

We deal with the *what* to send problem in this module by directly using the LLMs as a Message Generator with designed prompts, constructed from the components of Instruction Head, Goal Description, States Description, Action History, and Dialogue History. To better constrain LLMs' generated messages, we also add a note at the end of the prompt and append two seed messages at the beginning of the Dialogue History to elicit deserved effective communication behavior. The detailed prompt design is shown below:

**Instruction Head**   This part of the prompts is fixed for an environment, mainly consisting of the task instructions and environmental constraints.

**Goal Description**   For each task, the goal description is converted from $G = \{g_1, g_2, ..., g_k\}$ using a formal template.

**State Description**   For each step, the state description is converted from task progress, state of self, state of others, and semantic map retrieved from the Memory Module through a template.

**Action History**   The concatenation of the last $K$ actions (high-level plans) the agent took.

**Dialogue History**   The Concatenation of the last $D$ dialogues between agents including the messages the agent itself has sent.

To constrain the message generation of the LLMs, we add a note at the end of the prompt:

*Note: The generated message should be accurate, helpful, and brief. Do not generate repetitive messages.*

And append two seed messages at the beginning of the Dialogue History to elicit deserved effective communication behavior:

*Alice: "Hi, I'll let you know if I find any goal objects, finish any subgoals, and ask for your help when necessary."*

*Bob: "Thanks! I'll let you know if I find any goal objects, finish any subgoals, and ask for your help when necessary."*

## A.4  PLANNING MODULE

*CoELA* needs a strong Planning Module to make decisions on which action to take utilizing all available information gathered and stored so far to maximize cooperation efficiency.

While designing such a module from scratch consumes large human expert efforts and is nearly impossible to generalize, we utilize powerful LLMs directly as the Planning Module by first retrieving the related information from the Memory Module and converting them into text descriptions as in the Communication Module, then compile an Action List of all available high-level plans proposed according to the current state and the procedural knowledge stored for the LLMs to make the choice, which formalization makes it easier for the LLMs to concentrate on the reasoning and make an executable plan without any few-shot demonstrations easily, finally prompting the LLMs with current information and the proposed Action List to generate a high-level plan. We also use the zero-shot chain-of-thought prompting technique introduced by Kojima et al. (2022) to encourage the LLMs to carry out more reasoning before giving the final answer.

**Action List**  We compile all available actions regarding the current state into an Action List for the LLMs to select from. The multi-choice formalization makes it easier for the LLM to make an executable plan without any few-shot demonstrations. All available high-level plans on the TDW-MAT include

- go to room *
- explore current room
- go grasp target object/container *
- put holding objects into the holding container
- transport holding objects to the bed
- send a message: "*"

**Answer Extraction**  As shown in (Wei et al., 2022), chain-of-thought prompting can unleash the strong reasoning ability of the LLMs, we use the zero-shot chain-of-thought prompting technique introduced by (Kojima et al., 2022) to encourage the LLM to carry out more reasoning before giving the final answer.

## A.5  EXECUTION MODULE

To enable effective and generalized cooperation decision-making in different environments, we design an Execution Module to generate primitive actions to execute a given high-level plan robustly in a specific environment, allowing the Planning Module to be generalizable and focus more on solving the overall task with LLMs' rich world knowledge and strong reasoning ability. Practically, this design can also reduce the LLM inference time and is time-saving and economical. When facing a new environment with a different action space, only the procedural knowledge needs to be rewritten for *CoELA* to work. For rearrangement tasks, we mainly use an A-star-based planner to find the shortest path for navigation and robustly interact with the objects according to rules.

## A.6  A WORKING EXAMPLE ON TDW-MAT

To better understand our method, we present A working example of *CoELA* on one step in the TDW-MAT in Figure 7. *CoELA* receives an observation of $512\times512$ first-person view RGB image and Depth image from the environment, first uses the Perception Module implemented with Mask-RCNN to predict an instance segmentation mask, then builds 3D point clouds and extracts the states (positions, names, IDs, objects holding if agents) of the key objects including target objects, containers, and the agents, and builds a local occupancy map. The Memory Module uses the extracted states of the key objects and the local occupancy map to construct and update the semantic map, which is stored in Semantic Memory. The Memory Module also stores the task progress, the states

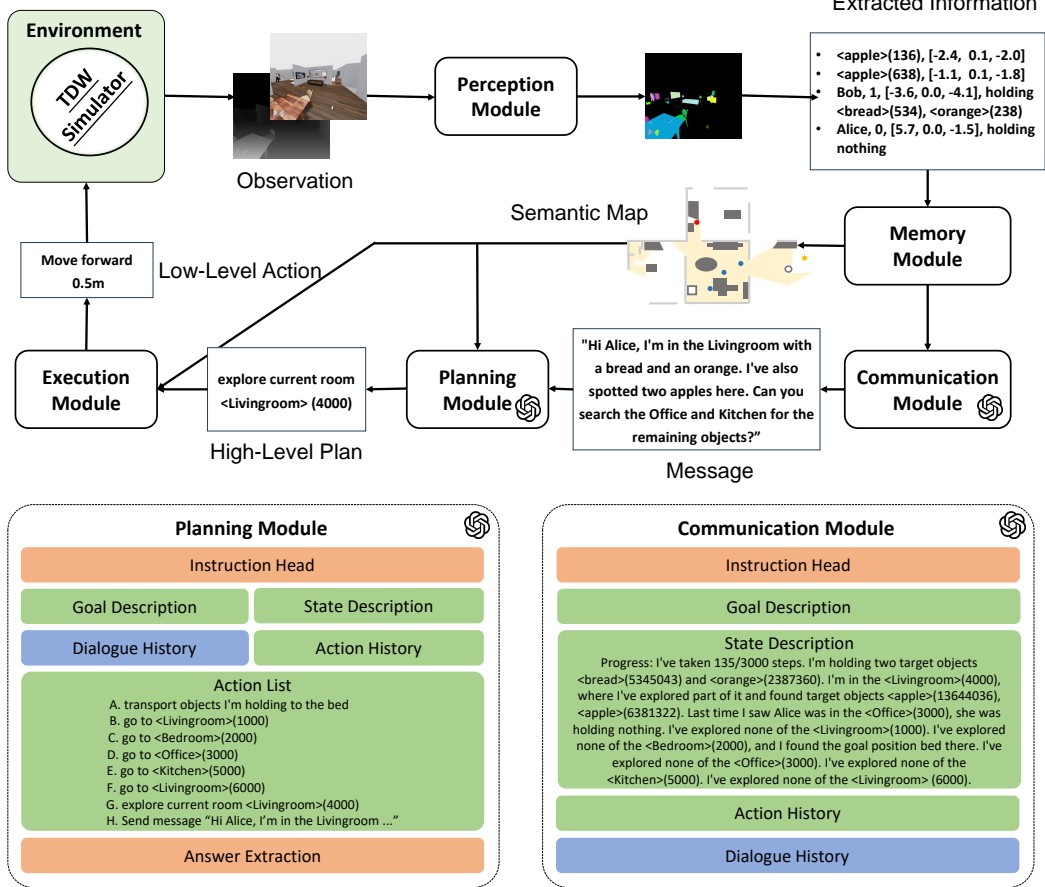

Figure 7: A working example on the TDW-MAT. The environment provides an observation of 512 * 512 first-person view RGB image and Depth image. The Perception Module takes these in, builds 3D point clouds, then extracts the states (positions, names, IDs, objects holding if agents) of the key objects including target objects, containers, and the agents, and builds a local occupancy map. The Memory Module uses the extracted states of the key objects and the local occupancy map to construct and update the semantic map, which is stored in Semantic Memory. The Memory Module also stores the task progress, the states of the agents in the Semantic memory, and the agent's action and dialogue history in the Episodic Memory, which are also updated when a message is received. The Communication Module converts the semantic map, task progress, and agents' states into textual State Description and concatenates it with the Instruction Head, Goal Description, Action History, and Dialogue History as the prompt to condition the LLM on current states and generate the message to be sent beforehand. The Planning Module similarly takes these inputs and converts them into a prompt with the addition of an Action List compiled with all available high-level plans including sending the message just generated, then taking advantage of the chain-of-thought prompting to decide on the high-level plan. The Execution Module first uses an A-Star-based planner to find the shortest path from the current location to the target location with the help of the semantic map if needed, then carry out the interaction required to finish the high-level plan.

of the agents in the Semantic memory, and the agent's action and dialogue history in the Episodic Memory, which are also updated when a message is received. The Communication Module converts the semantic map, task progress, and agents' states into textual State Description and concatenates it with the Instruction Head, Goal Description, Action History, and Dialogue History as the prompt to condition the LLM on current states and generate the message to be sent beforehand. The Planning Module similarly takes these inputs and converts them into a prompt with the addition of an Action List compiled with all available high-level plans including sending the message just generated, then taking advantage of the chain-of-thought prompting to decide on the high-level plan "explore current room <Livingroom> (4000)". The Execution Module then uses an A-Star-based planner to find the shortest path from the current location to the target location with the help of the semantic map and gives the low-level primitive action of "Move forward 0.5m", which is carried out in the environment and the new observation will be sent to the agents again.

# B ADDITIONAL DETAILS ON ENVIRONMENTS

## B.1 THREEDWORLD MULTI-AGENT TRANSPORT

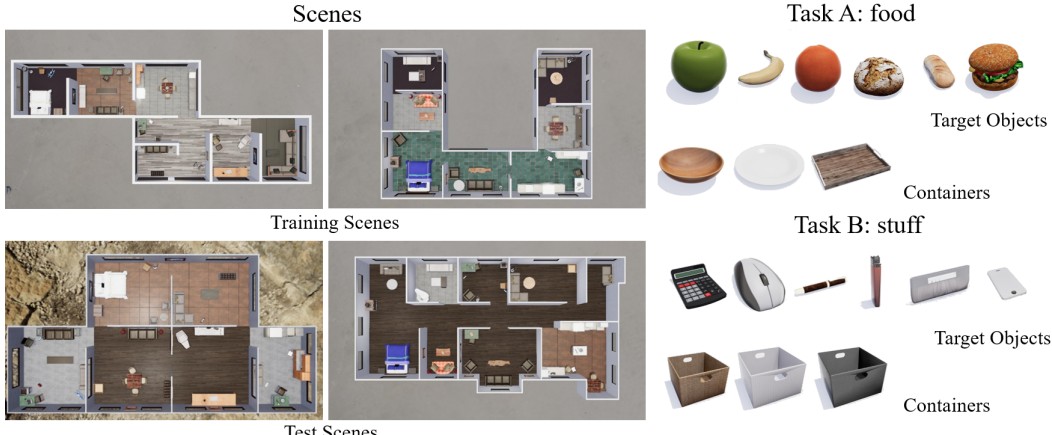

Figure 8: TDW-MAT scenes, target objects, and containers.

As an extension of the ThreeDWorld Transport Challenge(Gan et al., 2021), ThreeDWorld Multi-Agent Transport (TDW-MAT) supports multi-agent cooperation with natural language communication and includes more types of objects with more realistic placements. In the new challenge, we use the latest *replicant* humanoid provided by the TDW platform as an embodiment.

**Tasks** Two tasks are available in TDW-MAT: *food-transporting task* and *stuff-transporting task*. The two tasks have different types of target objects and containers. Figure 8 shows an overview of the two tasks: We create 4 floorplans and each of them has 3 layouts, where two floorplans are for the training set and another two are for the test set. The food-transporting task has 6 types of targets (apple, banana, orange, bread, loaf bread, and burger) and 3 containers (bowl, plate, and tea tray). In contrast, the stuff-transporting task has 6 different types of targets(calculator, mouse, pen, lighter, purse, and iPhone) and 3 containers (plastic basket, wood basket, and wicker basket). In each task, there are 10 target objects and 2 to 5 containers in total. Additionally, there are 4 types of rooms: living room, office, kitchen, and bedroom, and objects are placed in these rooms consistent with common sense. For example, food is more likely to be found in kitchens, while stuff is often in offices.

The agents are tasked to transport as many target objects as possible to the goal position with the help of containers as tools. One container can carry most three objects, and without containers, the agent can transport only two objects at a time. Agents need to transport target objects as much as possible within 3000 frames.

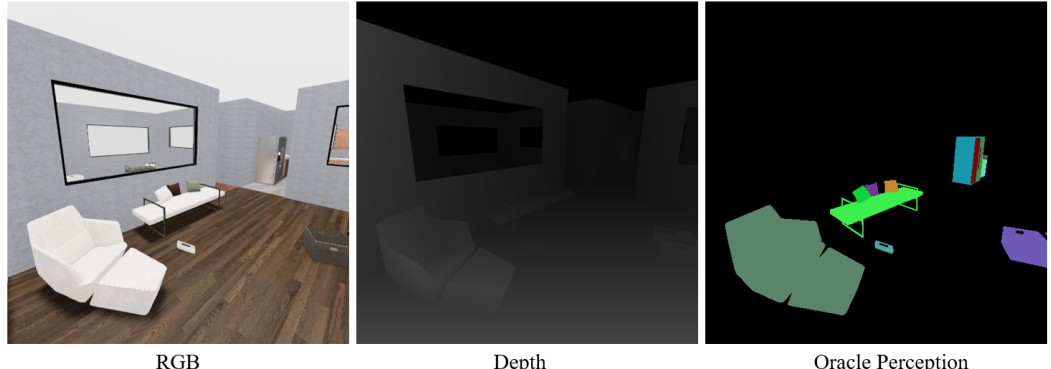

RGB      Depth      Oracle Perception

Figure 9: The RGB, depth, and oracle perception generated from the TDW-MAT environment.

**Observation Space** The embodied agent receives the egocentric RGB image and depth image as the main observation, as well as some auxiliary observations. Figure 9 is an example of an image generated from the TDW-MAT environment, and the detailed observation space is listed here:

- **RGB image:** the egocentric image comes from the camera facing forward, with screen size $512 \times 512$ and field of view 90;
- **Depth image:** the depth image has the same camera intrinsic parameters as the RGB image;
- **Oracle Perception (optional):** an image where each object id is mapped to a color and the camera intrinsic parameters are the same as the RGB image;
- **Agent position and rotation**: the agent's position and rotation in the simulation world;
- **Messages**: the messages sent by all the agents;

**Action Space** In TDW-MAT, there are 7 types of actions for agents to interact with the environment or communicate with each other. Each action takes several frames and the detailed action space is listed here:

- **Move forward**: move forward 0.5m;
- **Turn left**: turn left by 15 degrees;
- **Turn right**: turn right by 15 degrees;
- **Grasp**: grasp an object, only the agent is close to the object can he perform the action successfully. The object can be either a target or a container;
- **Put In**: put the target into the container, only the agent is holding a target in one hand and a container in another hand can he perform the action.
- **Drop**: drop the objects held in hand;
- **Send message**: Send a message to other agents. In each frame, no more than 500 characters can be sent.

### B.2 COMMUNICATIVE WATCH-AND-HELP

Communicative Watch-And-Help (C-WAH) is an extension of the Watch-And-Help challenge(Puig et al., 2021), which enables agents to send messages to each other. Sending messages, alongside other actions, takes one timestep and has an upper limit on message length.

**Tasks** Five types of tasks are available in C-WAH, named *Prepare afternoon tea*, *Wash dishes*, *Prepare a meal*, *Put groceries*, and *Set up a dinner table*. These tasks include a range of housework, and each task contains a few subgoals, which are described by predicates. A predicate is in "*ON/IN(x, y)*" format, that is, "*Put x ON/IN y*". The detailed descriptions of tasks are listed in Table 3.

The task goal is to satisfy all the given subgoals within 250 time steps, and the number of subgoals in each task ranges from 3 to 5.

| Task Name | Predicate Set |
|---|---|
| Prepare afternoon tea | ON(cupcake,coffeetable), ON(pudding,coffeetable), ON(apple,coffeetable), ON(juice,coffeetable), ON(wine,coffeetable) |
| Wash dishes | IN(plate,dishwasher), IN(fork,dishwasher) |
| Prepare a meal | ON(coffeepot,dinnertable),ON(cupcake,dinnertable), ON(pancake,dinnertable), ON(poundcake,dinnertable), ON(pudding,dinnertable), ON(apple,dinnertable), ON(juice,dinnertable), ON(wine,dinnertable) |
| Put groceries | IN(cupcake,fridge), IN(pancake,fridge), IN(poundcake,fridge), IN(pudding,fridge), IN(apple,fridge), IN(juice,fridge), IN(wine,fridge) |
| Set up a dinner table | ON(plate,dinnertable), ON(fork,dinnertable) |

Table 3: **Task description in C-WAH**. There are 5 types of tasks and each of them contains a few predicates.

**Observation Space**   C-WAH has two observation modes, named *Symbolic Observation* and *Visual Observation*. For *Symbolic Observation*, we followed the setting of the original Watch-And-Help challenge, one agent can receive all the object information in the same room as the agent, and the information includes location, status, name, relationship, etc.

For *Visual Observation*, agents can receive the egocentric RGB image and depth image, as well as some auxiliary observations. The detailed observation space is listed here:

- **RGB image:** the egocentric image comes from the camera facing forward, with screen size $256 \times 512$ and field of view 60;

- **Depth image:** the depth image has the same camera intrinsic parameters as the RGB image;

- **Oracle Perception:** it is an image where each object id is mapped to a color and the camera intrinsic parameters are the same as the RGB image;

- **Agent position:** the agent's position in the simulation world;

- **Messages**: the messages sent by all the agents.

**Action Space**   The action space is similar to that in the original Watch-And-Help Challenge, with a new action *sending message* added. The detailed action space is listed here:

- **Walk towards**: move to an object in the same room with the agents or a room;

- **Turn left**: turn left by 30 degrees;

- **Turn right**: turn right by 30 degrees;

- **Grasp**: grasp an object, only the agent is close to the object can he perform the action successfully;

- **Open**: Open a closed container, only the agent is close to the container can he perform the action successfully;

- **Close**: Close an open container, only the agent is close to the container can he perform the action successfully;

- **Put**: Put the held objects into an open container or onto a surface, only the agent is close to the target position can he perform the action successfully;

- **Send message**: Send a message to other agents. no more than 500 characters can be sent at a time.

## C  ADDITIONAL DETAILS ON EXPERIMENTS

### C.1  TRAINING DETAILS ON THE MULTI-AGENT TRANSFORMERS

**Multi-Agent-Transformer(MAT)** We adopt Multi-Agent-Transformer(MAT) (Wen et al., 2022), which regards MARL as a sequence modeling problem and applies a centralized decision transformer to generate actions.

The input of MAT contains two parts, the first part is a top-down semantic map with size $(12, 24)$ from the oracle perception. The map has 9 channels, implying whether the place is a free space/obstacle/wall/unexplored space/target object location/container location/goal location/my location/another agent's location, and the second part is the agent information(whether holds a container, holding object counts, etc.). The output of MAT is one of the following actions: explore, navigate to the nearest target object, navigate to the nearest container, and navigate to the goal place. Each action will last for up to 50 frames or the action is finished.

We train our RL agents for $2e5$ frames with the hidden layer dim $64$, learning rate $7e-4$, ppo epoch $10$ on training sets. After training, we test the RL agent on the test sets.

### C.2  ADDITIONAL DETAILS ON OTHER BASELINES

**Rule-based Hierarchical Planner (RHP)** We adopt the strong performing baseline from the original challenge, which is a Rule-based Hierarchical Planner with Frontier Exploration strategy, consisting of a rule-based high-level planner that selects one of the high-level plans from Exploration, Pick up an object, Pick up a container, and Place according to some human-defined rules and an A-star based planner to navigate with occupancy map and semantic map obtain and updated from the visual observation. The Frontier exploration strategy randomly samples a way-point from an unexplored area as a sub-goal for exploration.

**MCTS-based Hierarchical Planner (MHP)** We adopt the strongest baseline from the original Watch-And-Help Challenge, which is a Hierarchical Planner with a high-level planner based on MCTS and a low-level planner based on regression planning (RP). MHP infers the other's intention and adapts its subgoal accordingly based on the observation of the other agent.

### C.3  ADDITIONAL DETAILS ON *CoLLAMA*

We collected 2k trajectories from 10 episodes in the training set of TDW-MAT with GPT-4 driven *CoELA* and manually filtered 572 high-quality data with effective communication behavior and good reasoning trace towards collaborative decision-making. We use LoRA to fine-tune the LLAMA-2-13b-chat with a batch size of 384, a maximal sequence length of 2048, and a max learning rate of $4e^{-4}$ for 30 epochs (approximately 60 steps).

### C.4  ADDITIONAL QUALITATIVE ANALYSIS OF THE AGENT BEHAVIORS

***CoELA* exhibit efficient communication and effective cooperation behavior** To better understand the essential factors for effective cooperation, we conduct a qualitative analysis of the agents' behaviors exhibited in our experiments and identified several cooperative behaviors, as shown in Figure 3.

***CoELA* shares progress and information with others.** As shown in Figure 3abde, *CoELA* communicate with each other to share progress and intents, demonstrating **the Communication Module can handle the challenge of *what* to send**, harnessing the free dialogue generation ability from the LLMs.

***CoELA* knows when to request help and can respond to others' requests.** In Figure 3d, Bob finds a target object in the living room but his container is already full, so he shares this information and requests Alice to come here to help. Alice responds by going there and grabbing the objects. Similarly in Figure 3b, Alice responds to Bob's requests and questions. These examples show *CoELA* know when to request help and can understand others' requests and responses.

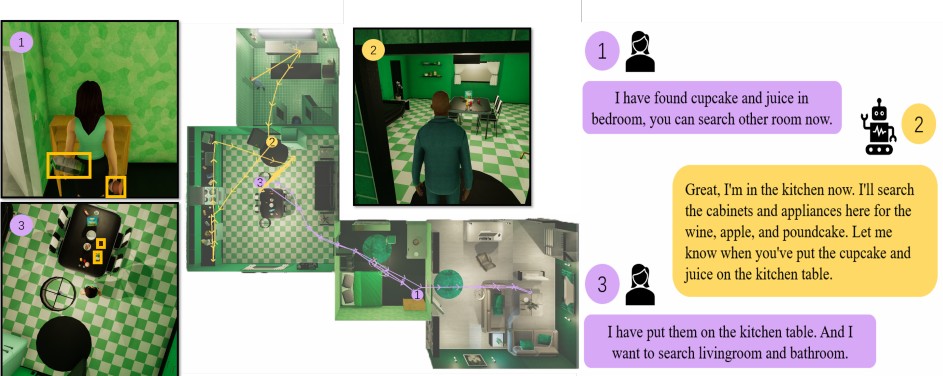

Figure 10: A qualitative example in Human + *CoELA* experiments, showcasing *CoELA* can communicate with Humans well and end up with a perfect division of the exploration trajectory.

*CoELA* **can adapt plans considering others.** In Figure 3a, Bob suggests a labor division of himself going to the kitchen while Alice checks the other rooms, but Alice suggests a better plan given her circumstances that she's already in the kitchen which Bob is not aware of before, and finally, Bob adapts his plan to cooperate with her.

*CoELA* **know when not to communicate.** In Figure 3c, though Bob receives Alice's suggestion of sharing any progress and has just found a plate, it's more efficient for him to grab the objects by himself and get the job done since this is the last goal object. He successfully reasons about this and chooses not to communicate to achieve higher efficiency. We also observed this behavior from humans when conducting the same task.

## C.5 ADDITIONAL DETAILS ON THE HUMAN EXPERIMENTS

We show an effective communication example in Figure 10, where the human first shares his progress with *CoELA* and suggests a labor division, *CoELA* understands and responds with its future plan as well, resulting in a perfect division of the exploration trajectory. These results imply promising futures for leveraging LLMs to build cooperative embodied agents that can successfully work with humans.

## D ADDITIONAL DISCUSSIONS

*CoELA* **is prone to cooperation**  Communication doesn't ensure consensus, and arguing back and forth can consume significant time, resulting in reduced efficiency. Interestingly though understandable, we did not observe such a phenomenon during our experiments. *CoELA* is prone to cooperation and coordinate plans without arguing back and forth which may be credited to LLMs trained to follow instructions and trust their cooperators. This behavior is beneficial for cooperation, though it may lead to less efficiency when the cooperator is malicious.

**Language Agents for Embodied Planning**  With the recent advance of Large Language Models, there has been work emerging to leverage LLMs to build powerful Embodied Agents. Huang et al. (2022a) used GPT-3 to generate high-level plans directly in a non-interactive way and used another smaller Language Model to translate the plan to available actions on virtualhome. Liang et al. (2022); Song et al. (2022) used codes or few-shot prompting to directly generate plans, Huang et al. (2022b) built an inner monologue with environment feedback to improve planning, Ahn et al. (2022) combined robotic affordances and LLMs for grounded instruction following. More recently, Park et al. (2023) built an agent society using LLMs augmented with memories in a sandbox environment to simulate human behavior. In contrast to the above, our work addresses a more *challenging* multi-agent cooperation problem, characterized by decentralized control, complex observations, **costly communication**, and **long-horizon multi-objective tasks**.

# E   EXAMPLE PROMPTS

We show an example prompt for the Planning Module on C-WAH in Table 4, and an example prompt for the Planning Module on TDW-MAT in Table 6.

Table 4: Example prompt for the Reasoning Module on C-WAH

---

**C-WAH Prompts**

```
I'm Alice. I'm in a hurry to finish the housework with my friend
Bob together. Given our shared goal, dialogue history, and my
progress and previous actions, please help me choose the best
available action to achieve the goal as soon as possible. Note
that I can hold two objects at a time and there are no costs for
holding objects. All objects are denoted as <name> (id), such as
<table> (712).
Goal: Find and put 1 wine, 1 pancake, 1 poundcake, 1 juice, 1
apple onto the <kitchentable> (130).
Progress: I'm holding nothing. I'm in the <kitchen>, where I found
<kitchentable> (130), <apple> (386), and unchecked containers
<kitchencabinet> (137), <kitchencabinet> (138), <kitchencabinet>
(139), <kitchencabinet> (140), <stove> (157), <dishwasher> (159),
<fridge> (165), <fridge> (166), <microwave> (167). I also see Bob
here in the <kitchen>, he is holding <apple> (387). I found an
unchecked container <bathroomcabinet> (25) in the bathroom. The
livingroom is unexplored. The bedroom is unexplored.
Dialogue history:
Alice: "Hi, I'll let you know if I find any goal objects and
finish any subgoals, and ask for your help when necessary."
Bob: "Thanks! I'll let you know if I find any goal objects and
finish any subgoals, and ask for your help when necessary."
Alice: "Bob, I'm in the kitchen with the <kitchentable> (130) and
found <apple> (386). Please check the living room and bedroom for
the other items. I will search the kitchen cabinets and
appliances."
Bob: "Alice, I found an <apple> (387) and I'm in the <livingroom>.
Can you search the kitchen for the other items? I'll check the
bathroom and bedroom next."
Previous actions: [goexplore] <bathroom> (11), [goexplore]
<kitchen> (56), [send_message], [gocheck] <kitchencabinet> (133),
[gocheck] <kitchencabinet> (134), [gocheck] <kitchencabinet>
(135), [gocheck] <kitchencabinet> (136)
Available actions:
A. [send_message] <"Bob, I see you have an <apple> (387). Please
put it on the <kitchentable> (130) and check the bedroom while I
search the remaining kitchen cabinets and appliances for the other
items.">
B. [goexplore] <livingroom> (198)
C. [goexplore] <bedroom> (294)
D. [gocheck] <kitchencabinet> (137)
E. [gocheck] <kitchencabinet> (138)
F. [gocheck] <kitchencabinet> (139)
G. [gocheck] <kitchencabinet> (140)
H. [gocheck] <stove> (157)
I. [gocheck] <dishwasher> (159)
J. [gocheck] <fridge> (165)
K. [gocheck] <fridge> (166)
L. [gocheck] <microwave> (167)
M. [gocheck] <bathroomcabinet> (25)
N. [gograb] <apple> (386)
Answer: Let's think step by step.
```

---

**LLM outputs**

```
First, we know that Bob is already holding an <apple> (387) and is
in the <livingroom>. We also know that we need to find and put 1
wine, 1 pancake, 1 poundcake, 1 juice, and 1 apple onto the
<kitchentable> (130). Since we are already in the <kitchen> and
there are several unchecked containers, the best course of action
is to continue searching the kitchen cabinets and appliances for
the remaining items.
So, the best available action for Alice is:
D. [gocheck] <kitchencabinet> (137)
```

Table 6: Example prompt for the Reasoning Module on TDW-MAT

---

**TDW-MAT Prompts**

I'm Alice. My friend Bob and I want to transport as many target
objects as possible to the bed with the help of containers within
3000 steps. I can hold two things at a time, and they can be
objects or containers. I can grasp containers and put objects into
them to hold more objects at a time. Given our shared goal,
dialogue history, my progress, and previous actions, please help
me choose the best available action to achieve the goal as soon as
possible. Note that a container can contain three objects, and
will be lost once transported to the bed. I can only put objects
into the container I hold after grasping it. All objects are
denoted as <name> (id), such as <table> (712). Actions take
several steps to finish. It may be costly to go to another room or
transport to the bed, use these actions sparingly.
Goal: Transport 3 pens, 1 lighter, 3 ipods, 2 purses, 1 key to the
bed.
Progress: I've taken 1313/3000 steps. We've already transported
<key> (3207585), <purse> (15433283), <ipod> (6544816), <purse>
(11543537), <pen> (12835254) to the bed. I'm holding nothing. I'm
in the <Bedroom> (2000), where I've explored all of it and found
the goal position bed. Last time I saw Bob was in the <Office>
(3000), he was holding nothing. I've explored part of the
<Livingroom> (1000), and I found a container <wicker_basket>
(13929244) there. I've explored none of the <Office> (3000), and I
found target objects <ipod> (2031829), <ipod> (16421571) there.
I've explored part of the <Livingroom> (4000). I've explored none
of the <Kitchen> (5000). I've explored all of the <Livingroom>
(6000), and I found a target object <pen> (1440084) there.
Dialogue history:
Alice: "Hi, I'll let you know if I find any target objects and
containers, finish any subgoals, and ask for your help when
necessary."
Bob: "Thanks! I'll let you know if I find any target objects and
containers, finish any subgoals, and ask for your help when
necessary."
Alice: "Bob, I suggest you transport the basket with key and purse
to the bed in the Bedroom (2000). Meanwhile, I'll go to the
Livingroom (6000) to grab the pen and the container from
Livingroom (1000) to collect more items. Let's meet at the bed
afterward to plan further."
Bob: "Alice, I'm heading to the Bedroom to transport the basket
with key and purse. After you grab the pen and container, let's
meet at the bed to plan our next steps. Let me know if you find
any other target objects."
Bob: "Alice, I just transported the key and purse to the bed.
Please grab the pen and container you found and head to the bed.
Let me know if you find any other target objects or containers on
your way."
Previous actions: go to <Bedroom> (2000) at step 396, send a
message at step 584, go grasp container <plastic_basket>
(15946891) at step 585, go grasp target object <ipod> (6544816) at
step 880, put <ipod> (6544816) into the container <plastic_basket>
(15946891) at step 889, go grasp target object <purse> (11543537)
at step 929, put <purse> (11543537) into the container
<plastic_basket> (15946891) at step 1033, go grasp target object
<pen> (12835254) at step 1057, put <pen> (12835254) into the
container <plastic_basket> (15946891) at step 1098, transport
objects I'm holding to the bed at step 1131

---

---

**TDW-MAT Prompts**

```
Available actions: (You can only choose the action in the list)
A. send a message: "Bob, I transported an ipod, purse, and pen to
the bed. I found a container <wicker_basket> (13929244) in the
<Livingroom> (1000) and target objects <ipod> (2031829) and <ipod>
(16421571) in the <Office> (3000). Please collect the ipods and
explore the <Office> for more objects. I'll collect the container
and pen from the <Livingroom> and meet you at the bed."
B. go grasp target object <pen> (1440084)
C. go grasp target object <ipod> (2031829)
D. go grasp target object <ipod> (16421571)
E. go grasp container <wicker_basket> (13929244)
F. go to <Livingroom> (1000)
G. go to <Office> (3000)
H. go to <Livingroom> (4000)
I. go to <Kitchen> (5000)
J. go to <Livingroom> (6000)
Answer: Let's think step by step.
```

---

**LLM outputs**

```
First, we need to collect the remaining target objects. You should
go to the <Livingroom> (6000) to grasp the <pen> (1440084) since
it's a target object. So, the best available action is:
B. go grasp target object <pen> (1440084)
```

## F  TEMPLATE LANGUAGE COMMUNICATION FOR MCTS-BASED HP AGENT

To enable the MCTS-based Hierarchical Planner to also leverage the new communication capability, we design three types of template language communication for it.

**Progress Sharing** $P$   An important motivation for communication is to share progress with others to update each other about the total progress of the task. So we design the first template communication to share progress.

When the agent carries out the action of *put*, which implies a new subgoal has been achieved by the agent, it will send a message such as:

 'P': 'I successfully put poundcake <383> on kitchentable <130>, and they are in kitchen <56>. '

When the agent receives such a message, it will process it and extract the sub-goal satisfied, and use it to update its inner tracking of the task progress, so avoiding taking an already satisfied sub-goal as a sub-goal again to better cooperate.

**Intent Sharing** $I$   Another important motivation for communication is to share intent with each other so that all the agents can plan coordinately together. So we design a template communication to share intent.

When the agent changes its sub-goal (practically, the Monte Carlo Tree Search High-Level Planner gives a new plan), it will tell the other agents its current sub-goal by sending a message such as:

'I': 'Now I want to put cutleryfork <369> in dishwasher <104>, and I have not found it yet. '

When the agent receives such a message, it will process it and extract the other agents' new sub-goal and update its belief about the others' intents, so it will not choose the same sub-goal with the others to avoid duplicate and improve efficiency.

**Belief Sharing** $B$    Sharing the scenes the agent just sees to the other agents can help them update their belief of the location of the object as well, and more importantly, this can help agents to build common ground on the belief of the objects to better cooperate together. So we also design a template communication to share beliefs.

When entering a new room, the agent will send all goal objects found or containers newly checked with no findings or target objects in it to others, such as:

'B': 'I found nothing is inside kitchencabinet <75>. nothing is inside kitchencabinet <76>. nothing is inside dishwasher <104>. nothing is inside cabinet <216>. cutleryfork <369>, cutleryfork <370> and plate <373> are inside kitchen <11>.'

When the agent receives such a message, it will process and extract the information maintained in the message to update its belief of the location distributions of the objects just as it has been seen by itself.

Also to be noticed, the agents may combine these three types of template communication to send one combined message at one time instead of multiple messages over several steps to improve efficiency.

