# OpenReview forum: "Building Cooperative Embodied Agents Modularly with Large Language Models"
_ICLR.cc/2024/Conference — ICLR 2024 poster_

### Official Review · Reviewer_xiUJ · 2023-10-19

**Soundness:** 2 fair
**Presentation:** 3 good
**Contribution:** 2 fair
**Rating:** 6
**Confidence:** 3

**Summary:**

This work proposes a framework called Cooperative Embodied Language Agent (CoELA), which explores the potential of Large Language Models (LLMs) for multi-agent communication. The framework is composed of five modules, among which the communication module helps the agent cooperate more effectively than traditional methods. Experiments conducted on TDW-MAT and C-WAH have demonstrated the effectiveness of CoELA when compared to strong planning-based methods, showcasing effective communication. Furthermore, the authors also explore the potential of using open LMs as LLMs, which is an impressive aspect of their work.

**Strengths:**

1. The comprehensive experiments demonstrate that the proposed method can efficiently cooperate with other agents, which is very impressive.
2. The presentation is clear.

**Weaknesses:**

1. Details regarding how CoELA cooperates with other agents are lacking. Sections 4 and 5 do not mention the mechanisms for cooperation between agents, such as how one MHP cooperates with another MHP or with CoELA.
2. Section 5.3.1 is missing the results of CoELA when driven by CoLLAMA.
3. The authors appear to be focused on exploring the potential of using Large Language Models (LLMs) in the Discrete Execution setting through communication. However, it's important to consider that communication may lead to challenges in certain scenarios, such as agents failing to reach a consensus. A discussion of such scenarios seems to be missing.

**Questions:**

1. How does the traditional MHP cooperate with MHP or CoELA?
2. How does CoELA handle the scenario where no consensus is reached? For example, Alice wants Bob to goto A and Bob wants Alice to goto B.

---

> ### Author Response · Authors · 2023-11-19
> **Response to Reviewer xiUJ**
>
> *We appreciate the positive and constructive comments from you! We address your questions in detail below and have updated our paper according to your suggestions.*
>
>
> > Q1: How does the traditional MHP cooperate with MHP or CoELA?
>
>
> Both agents take the same environment observation as input, and output environment-acceptable primitive actions. Despite the inner working mechanism being different, CoELA can still model the MHP agent's state and plan cooperatively. We follow the design of MHP in [1], where MHP infers the other's intention and adapts its subgoal accordingly based on the observation of the other agent. We've added these details to the revised paper.
>
>
> > Q2: How does CoELA handle the scenario where no consensus is reached? For example, Alice wants Bob to goto A and Bob wants Alice to goto B.
>
> Great question! Communication doesn't ensure consensus, and arguing back and forth can consume significant time, resulting in reduced efficiency. Interestingly though understandable, we did not observe such a phenomenon during our experiments. **CoELA is prone to cooperation** and coordinates plans without arguing back and forth which may be credited to **LLMs trained to follow instructions** and trust their cooperators. An interesting case where CoELA adapts plans considering others is shown in Figure 3a and Section C.4. Bob initially suggests a division of labor, but Alice disagrees and suggests another plan, Bob gives it a thought and accepts the new plan. This behavior is **beneficial for cooperation**, though it may lead to less efficiency when the cooperator is malicious. We've added this discussion to the paper as well.
>
> [1] Watch-and-help: A challenge for social perception and human-ai collaboration. ICLR21
>
> *We sincerely appreciate your comments. Please feel free to let us know if you have further questions.*
>
> Best,
> Authors

---

> > ### Comment · Reviewer_xiUJ · 2023-11-21
> >
> > Thanks for your reply! My concerns have been addressed, and I maintain my score at this stage.

---

### Official Review · Reviewer_iNRX · 2023-10-31

**Soundness:** 3 good
**Presentation:** 2 fair
**Contribution:** 3 good
**Rating:** 6
**Confidence:** 3

**Summary:**

This paper proposes a modular framework that integrates the Large Language Models to build Cooperative Embodied Language Agents CoELA, which focuses on the multi-agent setting with decentralized control, complex partial observation, costly communication and multi-objective tasks. Empirical experiments on C-WAH and TDW-MAT show that CoELA can achieve promising cooperative performance. Additional experiments with real humans demonstrate that CoELA can earn more trust and cooperate more effectively with humans.

**Strengths:**

1. This work proposes one feasible approach to building embodied agents with Large Language Models which seems to be sound. This demonstrates the prosiming potential of LLMs to build cooperative embodied agents and inspires future research well.
2. The experimental setup is relatively comprehensive, including cooperative evaluation  with AI agents and real humans. Besides, the discussion of the experimental results is also very interesting and thorough.
3. The discussions about failure cases and limitations are appreciated.

**Weaknesses:**

1. Despite the fact that the structure of the manuscript is organized well, the description of the method is relatively brief, with some details not sufficiently elaborated. For example, the manuscript states that, for the execution module, CoELA will utilize the procedure stored in its Memory Module to execute the high-level plan. However, it is unclear what form these procedures take and how they are obtained for a specific environment.
2. If I'm not mistaken, given a specific environment, CoELA needs to manually design/list the possible high-level plans, which may be a relatively tedious workload. Similar issues may arise when determining the possible high-level plans on each state and defining the procedure for each plan. This significantly influences the generality of the method and it may be challenging to implement these manual work in complex problems. Besides, listing the possible plans may significantly increase the prompt length, which might influence the performance.
3. The baselines in the experiments are relatively simple and heuristic. More baselines, especially methods designed for embodied agents, are recommended.

**Questions:**

1. How does CoELA determine the valid high-level plans at each state? Is it determined manually?
2. Have there been previous works using LLM for implementing embodied agents? Can more discussion or even experimental comparisons be made with these works? What are the core contributions and innovations of CoELA compared to them?

---

> ### Author Response · Authors · 2023-11-19
> **Response to Reviewer iNRX**
>
> *We appreciate your positive comments on our contributions, well-designed experiments, and thorough discussions! We address your questions in detail below and updated our paper according to your suggestions.*
>
> > Q1: How does CoELA determine the valid high-level plans at each state? Is it determined manually?
>
> CoELA proposes valid high-level plans **procedurally** given the current state and procedural knowledge retrieved from the memory module. We chose to organize them in a multi-choice format so the LLM can concentrate on the reasoning and make an executable plan **without any few-shot demonstrations**. More details are in the Appendix A.4
>
> > Q2: Have there been previous works using LLM for implementing embodied agents? Can more discussion or even experimental comparisons be made with these works? What are the core contributions and innovations of CoELA compared to them?
>
> Previous works using LLM for implementing embodied agents mainly focus on single-agent and simple tasks, while we address a more **challenging multi-agent cooperation problem** characterized by decentralized control, complex observations, costly communication, and long-horizon multi-objective tasks. To solve these new challenges, we design CoELA where we incorporate LLMs into a modular framework and enable **direct communication between each other**, which is both **effective** and **interpretable**. We've added a discussion section in the revised paper.
>
> *We sincerely appreciate your comments. Please feel free to let us know if you have further questions.*
>
> Best,
> Authors

---

> > ### Comment · Reviewer_iNRX · 2023-11-20
> > **Response to Authors**
> >
> > Thanks for your reply! My concerns have been addressed.

---

### Official Review · Reviewer_2bPF · 2023-10-31

**Soundness:** 2 fair
**Presentation:** 2 fair
**Contribution:** 2 fair
**Rating:** 6
**Confidence:** 4

**Summary:**

This paper presents an innovative approach to address multi-agent cooperation challenges in decentralized settings with costly communication and raw sensory observations. The authors introduce a modular framework integrating Large Language Models (LLMs), resulting in the Cooperative Embodied Language Agent (CoELA), capable of efficient planning, communication, and cooperation.

**Strengths:**

1. The paper introduces a unique integration of Large Language Models within a modular framework for decentralized multi-agent cooperation, addressing practical challenges in varied embodied environments.
2. Robust empirical support is provided through comprehensive experiments and a user study, showcasing the effectiveness of the approach and its positive impact on human-agent cooperation.
3. The paper is well-articulated and structured, offering clear insights and setting a strong foundation for future research in multi-agent cooperation with embodied agents.

**Weaknesses:**

1. This method assumes a skill library that is manually defined for a specific domain, i.e. execution module. However, this limits its applicability in other domains where predefined skill libraries are not available.
2. The pipeline appears to be quite complex and relies on several hand-defined modules, including perception modules, three memory modules, planning, and execution. Are all of these modules necessary? Conducting an ablation study would provide better understanding.
3. The experimental design lacks breadth as it only considers one scenario. Including evaluations across multiple scenarios would strengthen the findings.
4. There is some ambiguity regarding the execution module, memory module, and perception module details. Were they all designed by humans using language-based approaches?

**Questions:**

See in weakness.

---

> ### Author Response · Authors · 2023-11-19
> **Response to Reviewer 2bPF**
>
> *We sincerely thank the reviewer for the time to read our paper, and we appreciate your constructive comments! We address your concerns in detail below.*
>
> > Q1: This method assumes a skill library that is manually defined for a specific domain, i.e. execution module. However, this limits its applicability in other domains where predefined skill libraries are not available
>
> - Our low-level skills are obtained from motion planning, which is **common** in embodied AI benchmarks [1][2][3][4], given the focus is more on high-level reasoning and planning.
> - Our framework is **flexible** thanks to our **modular design**, and can be **easily adapted** to various low-level control policies, including reinforcement learning and trajectory optimization, offering a distinct advantage over end-to-end methodologies
>
> > Q2: Are all of these modules necessary? Conducting an ablation study would provide a better understanding.
>
> - Thanks for the question! We do **have an ablation study** on the memory, communication, planning, and execution modules in the paper to have a better understanding of the effectiveness of each module, and the results are shown in Figure 4 \(c\) and discussed in section 5.4.
> - The results show that **all modules contribute to the best performance** of CoELA when cooperating with other agents and humans.
>     - Without the Memory Module, CoELA needs nearly doubled steps to finish the task
>     - A strong LLM to drive the Planning and Communication module of CoELA is vital for better efficiency
>     - Without the Execution Module, the agent struggles to make low-level control directly and fails most of the task
> - It's not trivial to ablate on the perception module under the visual image observation setting, so we also conduct experiments with the oracle perception given, and the result is shown in Table 1. The performance gap between TDW-MAT and w/ Oracle Perception is rather close, **showing the Perception Model is effective** as well.
> - On the other hand, our modular framework is not restricted to the number of modules and can be easily extended to deal with all kinds of challenging tasks.
>
> > Q3: The experimental design lacks breadth as it only considers one scenario
>
> This might not be true. We actually strive to conduct comprehensive experiments **on the two most challenging embodied multi-agent benchmarks**.
> - Our experiments are conducted across **two simulation platforms** and different tasks with **varied observation and action space**, where C-WAH has **5 kinds of activities** and TDW_MAT features **12 different scenes**.
> - We experiment under both scenarios of **collaborating with AI agents** (section 5.3.1) and **with humans** (section 5.3.2).
> - The rearrangement task is a **canonical task** for evaluating embodied agents featuring various challenges including **perception**, **navigation** and **interaction**.[5]
> - We didn't notice any better options for testing our cooperative embodied agents in challenging environments.
> - Designing more suitable embodied multi-agent benchmarks for this field is also a valuable future avenue, though beyond the scope of this work.
>
> > Q4: There is some ambiguity regarding the execution module, memory module, and perception module details. Were they all designed by humans using language-based approaches?
>
> The reviewer may misunderstand some parts of our framework. Actually, in Appendix A, we provide more details regarding the framework.
> - The execution module, memory module, and perception module are designed to mimic human modular cognitive architectures and **not using a language-based approach**.
> - To help clear the ambiguity, we provide more details on the framework in Appendix A, with a figure demonstrating a working example on TDW-MAT.
> - The Perception Module utilizes a Mask-RCNN to obtain the instance segmentation mask from an ego-centric RGB image observation, and combines it with the depth image and the agent’s position to project each pixel into the 3D world coordinate to obtain a 3D voxel semantic map.
> - The Execution Module is based on an A-star-based planner to find the best path for navigation and robustly interact with the objects through primitive actions according to environmental rules, which is a common design for hierarchical planning in the field. [3] [4]
>
> [1] Retrospectives on the embodied ai workshop.
>
> [2] Do As I Can, Not As I Say: Grounding Language in Robotic Affordances. CoRL22
>
> [3] Watch-and-help: A challenge for social perception and human-ai collaboration. ICLR21
>
> [4] The threedworld transport challenge: A visually guided task-and-motion planning benchmark towards physically realistic embodied ai. ICRA22
>
> [5] Rearrangement: A Challenge for Embodied AI
>
> *We wish that our response has addressed your concerns, and turned your assessment to the positive side. If you have any more questions, please feel free to let us know. We appreciate your time and constructive suggestions! Thank you!*
>
> Best,
> Authors

---

> > ### Author Response · Authors · 2023-11-21
> > **Thank you for your review and we are looking forward to your feedback!**
> >
> > Dear reviewer 2bPF,
> >
> > We have provided a detailed response to address your concerns and here's a summary:
> >
> > - Assumes a skill library
> >   - Our framework is **flexible** thanks to our modular design, and can be **easily adapted** to various low-level control policies
> > - Are all of these modules necessary? Conducting an ablation study would provide a better understanding.
> >   - We do **have an ablation study** in the paper shown in Figure 4 (c) and discussed in section 5.4.
> >   - The results show that **all modules contribute** to the best performance of CoELA when cooperating with other agents and humans.
> >   - On the other hand, our modular framework is **not restricted to the number of modules** and can be easily extended to deal with all kinds of challenging tasks.
> > - The experimental design lacks breadth as it only considers one scenario
> >   - We strive to conduct comprehensive experiments on the **two most challenging embodied multi-agent benchmarks**
> >   - The rearrangement task is a **canonical** task for evaluating embodied agents featuring various challenges including perception, navigation, and interaction.
> > - There is some ambiguity regarding the execution module, memory module, and perception module details. Were they all designed by humans using language-based approaches?
> >   - The execution module, memory module, and perception module are designed to mimic human modular cognitive architectures and **not use a language-based approach**.
> >   - We provide more details on the framework in Appendix A, with a figure demonstrating a working example on TDW-MAT.
> >
> > Thanks again for your suggestion to strengthen this work. As the rebuttal period is ending soon, we wonder if our response answers your questions and addresses your concerns. If yes, would you kindly consider raising the score? Thanks again for your very constructive and insightful feedback!
> >
> > Best,
> >
> > Authors

---

> > > ### Comment · Reviewer_2bPF · 2023-11-22
> > >
> > > I would like to express my gratitude to the authors for providing clarifications.
> > >
> > > Upon carefully reviewing the author's response and considering other reviews, I have gained a better understanding of their work.
> > >
> > > Therefore, I am revising my score from 5 to 6.

---

### Official Review · Reviewer_tquX · 2023-11-01

**Soundness:** 3 good
**Presentation:** 4 excellent
**Contribution:** 4 excellent
**Rating:** 8
**Confidence:** 4

**Summary:**

The manuscript presents a comprehensive study on constructing cooperative embodied agents using Large Language Models (LLMs), aiming to address multi-agent collaboration in decentralized settings with challenges like raw sensory observations, costly communication, and multi-objective tasks. The authors introduce CoELA, a Cooperative Embodied Language Agent, which integrates the LLMs' capabilities with a modular framework encompassing perception, memory, execution, communication, and planning. The system's performance is evaluated in two embodied environments: C-WAH and TDW-MAT, demonstrating its ability to outperform planning-based methods, particularly when driven by GPT-4. The use of natural language for agent communication is highlighted as a significant advantage, fostering trust and effectiveness in human-agent interactions.

**Strengths:**

**Robust Motivation:** The paper is grounded in a strong and compelling motivation to enhance agent collaboration in complex environments. By addressing the need for agents to effectively communicate and plan their actions in a coordinated manner, the authors establish a solid foundation for their research, showcasing a clear understanding of the challenges and opportunities in the field.

**Carefully Designed Pipeline:** The system architecture demonstrates a thoughtful and meticulous design, integrating multiple modules and dual language models to manage both communication and planning. This comprehensive approach ensures that each aspect of the agent's interaction is given due consideration, resulting in a pipeline that is both balanced and well-reasoned. The deliberate inclusion of separate models for different functions reflects the authors' dedication to creating a system that is tailored to meet the specific demands of agent collaboration.

**Thorough Analysis and Discussion:** The paper excels in providing an in-depth analysis and discussion of the results, helping readers to fully grasp the implications and nuances of the study. The authors do not shy away from addressing the limitations of their work, offering a balanced view that adds credibility to their findings. This level of detail ensures that the paper serves not only as a presentation of the proposed framework but also as a valuable resource for future research, encouraging further investigation and innovation in the field of agent collaboration.

**Weaknesses:**

**Complex Model and System Design:** The architecture of the system is intricate, requiring each agent to manage five different modules and two distinct LLMs for handling communication and planning. This complexity can lead to instability in the LLM's performance, especially when processing lengthy textual inputs describing complicated scenarios. Furthermore, the challenge to maintain scalability becomes apparent as the addition of objects and details can potentially overwhelm the system, limiting its extendability. The intricate design also indirectly contributes to the limited utilization of spatial information and the difficulty in effective reasoning over low-level actions, as these scenarios would require even longer prompts.

**Effectiveness of Communication:** The effectiveness of communication within the system seems to be suboptimal according to the ablation study. Therefore, a natural question is whether there is a possible improvement for the communication module.
Personally, I am interested in the issues in Figure 5a. I am curious if Alice misinterprets Bob's actions due to an incorrect perception or what. Will bi-directed communication, such as Alice asking Bob if he has placed the object in the container when she needs to know, plus Bob telling Alice what he is doing at the beginning and the end of one mission, might serve as a straightforward solution to the problems, highlighting the need for a more responsive and interactive communication system.

By addressing these weaknesses and optimizing the system accordingly, there is potential to enhance the performance and scalability of the framework, leading to more effective agent collaboration.

**Questions:**

1. How to evaluate the communication cost? Is there any tradeoff study on communication cost and effectiveness?
2. How does the system perform in scenarios with an increased number of objects and more complex interactions, and what measures are in place to maintain scalability?
3. For turning left/right, what will happen if the object in interest is on the back of the agent? How is the visual-related textual input organized? Did the work try to use the oracle information of the whole environment, including objects and relations?

---

> ### Author Response · Authors · 2023-11-19
> **Response to Reviewer tquX**
>
> *We sincerely thank the reviewer for the time to read our paper, and for liking our contribution, method, and presentations.*
>
> > Q1: How to evaluate the communication cost? Is there any tradeoff study on communication cost and effectiveness?
>
> - The communication cost is evaluated in metrics *Average Steps* and *Transport Rate*.
> - Communication in our setting takes time as in real life, and therefore has an impact on the timesteps needed to carry out a task, which is reflected in the evaluation metric *Average Steps* and *Transport Rate*.
> - As shown in B.1 and B.2 Action Space, the agent can send at most 500 characters in each frame, which means a communication action of sending a message of 600 characters will cost the agent 2 frames. The agents need to **reason over the pros and cons of each step of communication on its own**.
> - An extreme case of the tradeoff study is where the communication takes no cost, then the agent can just send everything in their observation to the others and discuss their plan thoroughly before taking the next action, making the collaboration essentially less "de-centralized", and leading to a better efficiency regarding actions.
>
> > Q2: How does the system perform in scenarios with an increased number of objects and more complex interactions, and what measures are in place to maintain scalability?
>
> Our evaluated scenarios are **complex** compared to other embodied environments and the system can still perform well.
> - As shown in section B.1, there are **18 types of objects and containers of interest** in the **multi-room scenes**, and **6-7 actions to interact with**, making a rather complicated scenario in the field.
> - Across our experiments, the prompt uses **453** tokens on average and **591** tokens at max, which is rather **economical**, thanks to our deliberately designed memory module to store and organize the objects of interest **effectively**.
> - Considering extreme scenarios with too many objects of interest, our system can easily integrate some more deliberately designed filter modules such as those designed in [1] for better scalability, thanks to the modular design of our system.
>
> > Q3: For turning left/right, what will happen if the object in interest is on the back of the agent? How is the visual-related textual input organized? Did the work try to use the oracle information of the whole environment, including objects and relations?
>
> - We have both results from  Oracle Perception where the instance segmentation is given by the environment and Learned Perception where only  ego-centric RGBD image from the agent's camera is given and a neural perception model is deployed to segment the objects.
> - Under both scenarios, there exists the **partial observation challenge**, if the object in interest is out of the agent's view, such as on the back of the agent, the agent need to actively explore the environment to find it, leading to a **navigation and exploration challenge**.
> - Partial observation is a key feature in our challenging, making effective cooperation more challenging and lead to our key insights. We donot assume the oracle information of the whole environment.
>
> [1] Plan, Eliminate, and Track — Language Models are Good Teachers for Embodied Agents
>
> *We sincerely appreciate your detailed and constructive comments. Please feel free to let us know if you have further questions.*
>
> Best,
> Authors

---

> > ### Comment · Reviewer_tquX · 2023-11-22
> > **Post-Rebuttal Comment**
> >
> > Thank you for your response. My concern is addressed.

---

### Author Response · Authors · 2023-11-19
**General Response to All Reviewers and ACs**

We thank all the reviewers and ACs for their time and efforts in reviewing our paper and giving insightful comments. In addition to the detailed response to specific reviewers, here we would like to highlight our contributions and clarify some concerns.

**1.Our Contributions**

We are glad to find that the reviewers have acknowledged our following contributions:

- **Promising idea of incorporating LLMs into modular frameworks to enhance agent collaboration in challenging environments**

    - Carefully Designed Pipeline that is both balanced and well-reasoned [tquX]
    - a unique integration of Large Language Models within a modular framework for decentralized multi-agent cooperation, addressing practical challenges [2bPF]
    - demonstrates the promising potential of LLMs to build cooperative embodied agents [iNRX]
    - the proposed method can efficiently cooperate with other agents, which is very impressive [xiUJ]
- **Comprehensive experiments design and in-depth analysis**
    - The paper excels in providing an in-depth analysis and discussion of the results [tquX]
    - Robust empirical support is provided through comprehensive experiments and a user study [2bPF]
    - the discussion of the experimental results is also very interesting and thorough [iNRX]
- **Valuable insights for future research**
    - a valuable resource for future research, encouraging further investigation and innovation in the field of agent collaboration [tquX]
    - offering clear insights and setting a strong foundation for future research in multi-agent cooperation with embodied agents [2bPF]
    - inspires future research well [iNRX]
    - the authors also explore the potential of using open LMs as LLMs, which is an impressive aspect of their work. [xiUJ]

**2.Clarifications**

- **The complex design**
    - The framework is **deliberately** designed to incorporate LLMs into **modular** frameworks mimicking human cognitive architecture to deal with challenging decentralized multi-agent cooperation scenarios.
    - Experiments on two challenging embodied AI benchmarks show our modular framework is **effective** and can be **adapted** to different observation and action spaces.
    - The ablation study and in-depth analysis in the paper show **all modules contribute to the best performance** of CoELA when cooperating with other agents and humans.

- **Communication Efficiency**
    - Communication in our setting takes time as in real life, so the agents need to **reason over the pros and cons of each step of communication on its own**.
    -  Our designed framework can show **effective communication behaviors** under the challenging setting, **especially when cooperating with humans**.
    -  This setting also leads to many interesting scenarios, such as where agents fail to reach a consensus as reviewer xiUJ suggested. We analyze the agent's behaviors exhibited in the experiments and show CoELA are **prone to cooperate** and coordinate plans without arguing back and forth which can be credited to LLMs trained to follow instructions and trust their cooperators.

- **The scalability of the framework**
    - Our evaluated scenarios are **complex** as there are **18 types of objects and containers of interest** in the **multi-room scenes**, and **6-7 actions to interact with**.
    - Across our experiments, the prompt uses **453** tokens on average and **591** tokens at max, which is rather **economical**, thanks to our deliberately designed memory module to store and organize the objects of interest **effectively**.
    - Considering even more complex scenarios with too many objects of interest, our framework can easily integrate some more deliberately designed filter modules for better scalability, thanks to the **modular design** of our system.

- **Predefined skill library**
    - Our low-level skills are obtained from motion planning, which is common in embodied AI benchmarks, given the focus is more on high-level reasoning and planning.
    - Our framework is **flexible** thanks to our **modular design**, and can be **easily adapted** to various low-level control policies, including reinforcement learning and trajectory optimization, offering a distinct advantage over end-to-end methodologies.

We hope our detailed responses below convincingly address all reviewers’ questions.

**Revision Summary**

- We add an additional discussion paragraph in Appendix D to discuss in depth about the related work of using LLMs for embodied planning as Reviewer **iNRX** suggested.
- We add more details about the cooperation mechanisms in Appendix C.2 as Reviewer **xiUJ** suggested.
- We add an additional discussion paragraph about CoELA's communication behavior in Appendix D as Reviewer **xiUJ** suggested.

---

### Author Response · Authors · 2023-11-21
**Thank you and we are looking forward to your post-rebuttal feedback!**

Dear AC and all reviewers:

Thanks again for all the insightful comments and advice, which helped us improve the paper's quality and clarity.

The discussion phase has been on for several days and we are still waiting for the post-rebuttal responses.

We would love to convince you of the merits of the paper. Please do not hesitate to let us know if there are any additional experiments or clarification that we can offer to make the paper better. We appreciate your comments and advice.

Best,

Author

---

### Meta-Review · Area_Chair_iVeh · 2023-12-17

**Metareview:**

The authors tackle the problem of multi-agent cooperation focusing on the creation of a modular approach to agent construction that has explicit modules for every aspect of planning including communication. Overall reviewers were happy with the work though admit that it is a rather complex piece to evaluate.  They appreciate the modular framework, the experiments design and analysis, and the use of open LLMs.  Importantly, the work does include ablations to make the contributions of all modules clear in performance.

**Justification For Why Not Higher Score:**

The reviewers are positive yet overall still cautious.  There are likely several reasons for this -- from the impact of each model/module to the complexity of experimental conditions, choices of environment.  It is difficult to fully disentangle the impacts of each decision/component.

**Justification For Why Not Lower Score:**

Everyone is in agreement that the motivation is clear and that this is an interesting/important area for the community to tackle.  Nobody votes for rejection.

---

### Decision · Program_Chairs · 2024-01-16

Accept (poster)